# Regret Bounds for Gaussian-Process Optimization in Large Domains

**Manuel Wüthrich**[*]
MPI for Intelligent Systems
Tübingen, Germany

**Bernhard Schölkopf**
MPI for Intelligent Systems
Tübingen, Germany

**Andreas Krause**
ETH Zurich
Zurich, Switzerland

## Abstract

The goal of this paper is to characterize Gaussian-Process optimization in the setting where the function domain is large relative to the number of admissible function evaluations, i.e., where it is impossible to find the global optimum. We provide upper bounds on the suboptimality (Bayesian simple regret) of the solution found by optimization strategies that are closely related to the widely used expected improvement (EI) and upper confidence bound (UCB) algorithms. These regret bounds illuminate the relationship between the number of evaluations, the domain size (i.e. cardinality of finite domains / Lipschitz constant of the covariance function in continuous domains), and the optimality of the retrieved function value. In particular, we show that even when the number of evaluations is far too small to find the global optimum, we can find nontrivial function values (e.g. values that achieve a certain ratio with the optimal value).

## 1 Introduction

In practice, nonconvex, gradient-free optimization problems arise frequently, for instance when tuning the parameters of an algorithm or optimizing the controller of a physical system (see e.g. Shahriari et al. [2016]) for more concrete examples). Different versions of this problem have been addressed e.g. in the multi-armed-bandit and the Bayesian-optimization literature (which we review below). However, the situation where the number of admissible function evaluations is too small to identify the global optimum has so far received little attention, despite arising frequently in practice. For instance, when optimizing the hyper-parameters of a machine-learning method or the controller-parameters of a physical system, it is typically impossible to explore the domain exhaustively. We take first steps towards a better understanding of this setting through a theoretical analysis based on the adaptive-submodularity framework by Golovin and Krause [2011]. It is not the purpose of the present paper to propose a novel algorithm with better performance (we modify the EI and UCB strategies merely to facilitate the proof), but rather to gain an understanding of how GP-optimization performs in the aforementioned setting, as a function of the number of evaluations and the domain size.

As is done typically, we model the uncertainty about the underlying function as a Gaussian Process (GP). The question is how close we can get to the global optimum with $T$ function evaluations, where $T$ is small relative to the domain of the function (i.e., it is impossible to identify the global optimum with high confidence with only $T$ evaluations). As a performance measure, we use the Bayesian simple regret, i.e., the expected difference between the optimal function value and the value attained by the algorithm. For the discrete case with domain size $N$, we derive a problem-independent regret bound (i.e., it only depends on $T$, $N$ and is worst-case in terms of the GP prior) for two optimization algorithms that are closely related to the well-known expected improvement (EI, Jones et al. [1998]) and the upper confidence bound (UCB, Auer et al. [2002], Srinivas et al.

---

[*]`manuel.wuthrich@pm.me`

35th Conference on Neural Information Processing Systems (NeurIPS 2021).

[2010]) methods. In contrast to related work, our bounds are non-vacuous even when we can only explore a small fraction of the function. We extend this result to continuous domains and show that the resulting bound scales better with the size of the function domain (equivalently, the Lipschitz constant of the covariance function) than related work (Grünewälder et al. [2010]).

## 1.1 Related Work

The multi-armed bandit literature is closest to the present paper. In the multi-armed bandit problem, the agent faces a row of $N$ slot machines (also called one-armed bandits, hence the name). The agent can decide at each of $T$ rounds which lever (arm) to pull, with the objective of maximizing the payoff. This problem was originally proposed in Robbins [1952] to study the trade-off between exploration and exploitation in a generic and principled way. Since then, this problem has been studied extensively and important theoretical results have been established.

Each of the $N$ arms has an associated reward distribution, with a fixed but unknown mean $F_n$. When pulling arm $n$ at round $t$, we obtain a payoff $Y_t := F_n + E_{n,t}$ where the noise is zero mean $\mathbb{E}[E_{n,t}] = 0$, independent across time steps and arms, and identically distributed for a given arm across time steps. Let us denote the arm pulled at time $t$ by $A_t$. Performance is typically analyzed in terms of the regret, which is defined as the difference between the mean payoff of the optimal arm and the one pulled at time $t$

$$R_t := \max_{n \in [N]} F_n - F_{A_t}. \tag{1}$$

Here, we are interested in the setting where $E_{n,t}$ is small or even zero and we are allowed to pull only few arms $T < N$. We classify related work according to the information about the mean $F_n$ and the noise $E_{n,t}$ that is available to the agent.

### 1.1.1 No Prior Information

Traditionally, most authors considered the case where only very basic information about the payoff $F_n + E_{n,t}$ is available to the agent, e.g. that its distribution is a member of a given family of distributions or that it takes values in an interval, typically $[0, 1]$.

In a seminal paper, Lai and Robbins [1985] showed that the cumulative regret $\sum_{t \in [T]} R_t$ grows at least logarithmically in $T$ and proposed a policy that achieves this rate asymptotically. Later, Auer et al. [2002] proposed an upper confidence bound (UCB) strategy and proved that it achieves logarithmic cumulative regret in $T$ uniformly in time, not just asymptotically. Since then, many related results have been obtained for UCB and other strategies, such as Thompson sampling Agrawal and Goyal [2011], Kaufmann et al. [2012]. A number of similar results have also been obtained for the objective of best arm identification Madani et al. [2004], Bubeck et al. [2009], Audibert and Bubeck [2010], where we do not care about the cumulative regret, but only about the lowest regret attained.

However, all of these bounds are nontrivial only when the number of plays is larger than the number of arms $T > N$. This is not surprising, since no algorithm can be guaranteed to perform well with a low number of plays with such limited information. To see this, consider an example where $F_i = 1$ and $F_n = 0 \ \forall n \neq i$. Since the agent has no prior information about which is the right arm $i$, the best it can do is to randomly try out one after the other. Hence it is clear that to obtain meaningful bounds for $T < N$ we need more prior knowledge about the reward distributions of the arms.

### 1.1.2 Structural Prior Information

In Dani et al. [2008] the authors consider the problem where the mean rewards at each bandit are a linear function of an unknown, $K$-dimensional vector. However, similarly to the work above, the difficulty addressed in this paper is mainly the noise, i.e. the fact that the same arm does not always yield the same reward. For $E_{n,t} = 0$, the $\mathcal{O}(K)$ cumulative regret bounds derived in this paper are trivial, since with $\mathcal{O}(K)$ evaluations we can simply identify the linear function.

### 1.1.3 Gaussian Prior

More closely related to our work, a number of authors have considered Gaussian priors. Bull [2011] for instance provides asymptotic convergence rates for the EI strategy with Gaussian-process priors on the unknown function.

Srinivas et al. [2010] propose a UCB algorithm for the Gaussian-process setting. The authors give finite-time bounds on the cumulative regret. However, these bounds are intended for a different setting than the one we consider here: 1) They allow for noisy observations and 2) they are only meaningful if we are allowed to make a sufficient number of evaluations to attain low uncertainty over the entire GP. Please see Appendix B for more details on the second point.

Russo and Van Roy [2014] use a similar analysis to derive bounds for Thompson sampling, which are therefore subject to similar limitations. In contrast, the bounds in Russo and Van Roy [2016] do not depend on the entropy of the entire GP, but rather on the entropy of the optimal action. However, our goal here is to derive bounds that are meaningful even when the optimal action cannot be found, i.e. its entropy remains large.

De Freitas et al. [2012] complement the work in [Srinivas et al., 2010] by providing regret bounds for the setting where the function can be observed without noise. However, these are asymptotic and therefore not applicable to the setting we have in mind.

Similarly to the present paper, Grünewälder et al. [2010] analyze the Bayesian simple regret of GP optimization. They provide a lower and an upper bound on the regret of the optimal policy for GPs on continuous domains with covariance functions that satisfy a continuity assumption. Here, we build on this work and derive a bound with an improved dependence on the Lipschitz constant of the covariance function, i.e., our bound scales better with decreasing length-scales (and, equivalently, larger domains). Unlike [Grünewälder et al., 2010], we also consider GPs on finite domains without any restrictions on the covariance.

### 1.1.4 Adaptive Submodularity

The adaptive-submodularity framework of Golovin and Krause [2011] is in principle well suited for the kind of analysis we are interested in. However, we will show that the problem at hand is not adaptively submodular, but our proof is inspired by that framework.

## 2 Problem Definition

### 2.1 The Finite Case

In this section we specify the type of bandit problem we are interested in more formally. The goal is to learn about a function with domain $\mathcal{A} = [N]$ (we will use $[N]$ to denote the set $\{1, ..., N\}$) and co-domain $\mathbb{R}$. We represent the function as a sequence $F = (F_n)_{n \in [N]}$. Our prior belief about the function $F$ is assumed to be Gaussian

$$F \sim \mathcal{N}(\mu, \Sigma). \tag{2}$$

At each of the $T$ iterations, we pick an action (arm) $A_t$ from $[N]$ at which we evaluate the function. After each action, an observation $Y_t \in \mathbb{R}$ is returned to the agent

$$Y_t := F_{A_t}. \tag{3}$$

Note that here we restrict ourselves to the case where the function can be evaluated without noise.

For convenience, we introduce some additional random variables, based on which the optimization algorithm will pick where to evaluate the function next. We denote the posterior mean and covariance at time $t$ by

$$M_t := \mathbb{E}[F | A_{:t}, Y_{:t}] \tag{4}$$

$$C_t := \mathbb{COV}[F | A_{:t}, Y_{:t}]. \tag{5}$$

In addition, we will need the maximum and minimum observations up to time $t$

$$\hat{Y}_t := \max_{k \in [t]} Y_k \quad \forall t \in \{1, .., T\} \tag{6}$$

$$\check{Y}_t := \min_{k \in [t]} Y_k \quad \forall t \in \{1, .., T\}. \tag{7}$$

Furthermore, we will make statements about the difference between the smallest and the largest observed value

$$\hat{\check{Y}}_t := \hat{Y}_t - \check{Y}_t \quad \forall t \in \{1, .., T\}. \tag{8}$$

Finally, for notational convenience we define

$$\hat{Y}, \check{Y}, \hat{\check{Y}} := \hat{Y}_T, \check{Y}_T, \hat{\check{Y}}_T. \tag{9}$$

Analogously, let us define the function minimum $\check{F} := \min_{n \in [N]} F_n$, maximum $\hat{F} := \max_{n \in [N]} F_n$ and difference $\hat{\check{F}} := \hat{F} - \check{F}$.

### 2.1.1 Problem Instances

A problem instance is defined by the tuple $(N, T, \mu, \Sigma)$, i.e. the domain size $N \in \mathbb{N}_{>0}$, the number of rounds $T \in \mathbb{N}_{>0}$ and the prior (2) with mean $\mu \in \mathbb{R}^N$ and covariance $\Sigma \in \mathbb{S}_+^n$, where we use $\mathbb{S}_+^n$ to denote the set of positive semidefinite matrices of size $n$.

## 2.2 The Continuous Case

The definitions in the continuous case are analogous. A problem instance here is defined by $(\mathcal{A}, T, \mu, k)$, i.e. the function domain $\mathcal{A}$, the number of rounds $T \in \mathbb{N}_{>0}$, a mean function $\mu : \mathcal{A} \to \mathbb{R}$ and a positive semi-definite kernel $k : \mathcal{A}^2 \to \mathbb{R}$.

# 3 Results

In this section we provide bounds on the Bayesian simple regret that hold for the two different optimization algorithms we describe in the following.

## 3.1 Optimization Algorithms

Two of the most widely used GP-optimization algorithms are the expected improvement (EI) (Jones et al. [1998]) and the upper confidence bound (UCB) (Auer et al. [2002], Srinivas et al. [2010]) methods. In the following, we define two optimization policies that are closely related:

**Definition 1.** *The expected improvement Bull [2011] is defined as*

$$\mathrm{ei}(\tau) := \int_{-\infty}^{\infty} \max\{x - \tau, 0\} \mathcal{N}(x) dx = \mathcal{N}(\tau) - \tau \Phi^c(\tau) \tag{10}$$

*where $\mathcal{N}$ is the standard normal density function and $\Phi^c$ is the complementary cumulative density function of a standard normal distribution. Furthermore, we use the notation*

$$\mathrm{ei}(\tau | \mu, \sigma) = \sigma \, \mathrm{ei}\left(\frac{\tau - \mu}{\sigma}\right) = \int_{-\infty}^{\infty} \max\{x - \tau, 0\} \mathcal{N}(x | \mu, \sigma) dx. \tag{11}$$

**Definition 2** (EI2). *An agent follows the EI2 strategy when it picks its actions according to*

$$A_{t+1} = \underset{n \in [N]}{\mathrm{argmax}} \max\left\{ \mathrm{ei}\left(\hat{Y}_t | M_t^n, \sqrt{C_t^{nn}}\right), \mathrm{ei}\left(-\check{Y}_t | -M_t^n, \sqrt{C_t^{nn}}\right) \right\} \tag{12}$$

*with the expected improvement* ei *as defined in Definition 1.*

**Definition 3** (UCB2). *An agent follows the UCB2 strategy when it picks its actions according to*

$$A_{t+1} = \underset{n \in [N]}{\mathrm{argmax}} \max\left\{ -\hat{Y}_t + M_t^n + \sqrt{C_t^{nn} 2 \log N}, \check{Y}_t - M_t^n + \sqrt{C_t^{nn} 2 \log N} \right\}. \tag{13}$$

The main difference to the standard versions of these methods (see Definition 6 and Definition 7) is that the algorithms here are symmetrical in the sense that they are invariant to flipping the sign of the GP. They maximize *and* minimize at the same time by picking the point which we expect to either increase the observed maximum *or* decrease the observed minimum the most. This symmetry is important for our proof, whether the same bound also holds for following the standard, one-sided EI or UCB strategies is an open question, we discuss this point in detail in Section 4.

## 3.2 Upper Bound on the Bayesian Simple Regret for the Extremization Problem

Here, we provide a regret bound for function extremization (i.e. the goal is to find both the minimum and the maximum) from which the regret bound for function maximization will follow straightforwardly. Note that the bound is problem-independent in the sense that it does not depend on the prior $(\mu, \Sigma)$.

**Theorem 1.** *For any instance $(N, T, \mu, \Sigma)$ of the problem defined in Section 2.1 with $N \geq T \geq 500$, if we follow either the EI2 (Definition 2) or the UCB2 (Definition 3) strategy, we have*

$$\frac{\mathbb{E}\left[\hat{\hat{F}}\right] - \mathbb{E}\left[\hat{\check{Y}}\right]}{\mathbb{E}\left[\hat{\hat{F}}\right]} \leq 1 - \left(1 - T^{-\frac{1}{2\sqrt{\pi}}}\right) \sqrt{\frac{\log(T) - \log\left(3 \log^{\frac{3}{2}}(T)\right)}{\log(N)}}. \tag{14}$$

*This guarantees that the expected difference between the maximum and minimum retrieved function value achieves a certain ratio with respect to the expected difference between the global maximum and the global minimum.*

*Proof.* The full proof can be found in the supplementary material in Appendix G, here we only give an outline. The proof is inspired by the adaptive submodularity framework by Golovin and Krause [2011]. The problem at hand can be understood as finding the policy (optimization algorithm) which maximizes an expected utility. Finding this optimal policy would require solving a partially observable Markov decision process (POMDP), which is intractable in most relevant situations. Instead, a common approach is to use a greedy policy which maximizes the expected single-step increase in utility at each time step $t$, which is in our case

$$A_{t+1} = \underset{a}{\operatorname{argmax}} \left( \mathbb{E}\left[\hat{Y}_{t+1} - \check{Y}_{t+1} | A_{t+1} = a, A_{:t}, Y_{:t}\right] - \mathbb{E}\left[\hat{Y}_t - \check{Y}_t | A_{:t}, Y_{:t}\right] \right). \tag{15}$$

This corresponds to the EI2 (Definition 2) strategy. The task now is to show that the greedy policy will not perform much worse than the optimal policy. Golovin and Krause [2011] show that this holds for problems that are adaptively submodular (among some other conditions). In the present problem, we can roughly translate this condition to: The progress we make at a given time step has to be proportional to how far the current best value is from the global optimum. While our problem is not adaptively submodular (see Appendix A), we show that a similar condition holds (which leads to similar guarantees). □

## 3.3 Upper and Lower Bound on the Bayesian Simple Regret for the Maximization Problem

For maximization, the goal is to minimize the Bayesian simple regret, i.e. the expected difference between the globally maximal function value and the best value found by the optimization algorithm

$$\mathbb{E}[\hat{F}] - \mathbb{E}[\hat{Y}]. \tag{16}$$

As is often done in the literature (see e.g. Srinivas et al. [2010]), we restrict ourselves here to centered GPs (i.e. zero prior mean; naturally, the mean will change during the optimization). To be invariant to scaling of the prior distribution, we normalize the regret with the expected global maximum:

$$\text{normreg} := \frac{\mathbb{E}\left[\hat{F}\right] - \mathbb{E}\left[\hat{Y}\right]}{\mathbb{E}\left[\hat{F}\right]}. \tag{17}$$

### 3.3.1 Upper Regret Bound for the EI2 and UCB2 Policies

We obtain the following upper bound on this normalized Bayesian simple regret:

**Corollary 1.** *For any instance $(N, T, \mu, \Sigma)$ of the problem defined in Section 2.1 with zero mean $\mu_n = 0 \; \forall n$ and $N \geq T \geq 500$, if we follow either the EI2 (Definition 2) or the UCB2 (Definition 3) strategy, we have*

$$\text{normreg} \leq 1 - \left(1 - T^{-\frac{1}{2\sqrt{\pi}}}\right) \sqrt{\frac{\log(T) - \log\left(3 \log^{\frac{3}{2}}(T)\right)}{\log(N)}}. \tag{18}$$

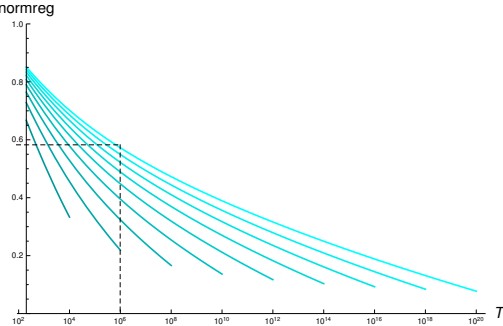
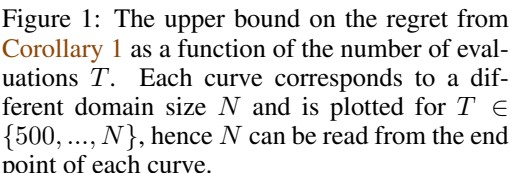
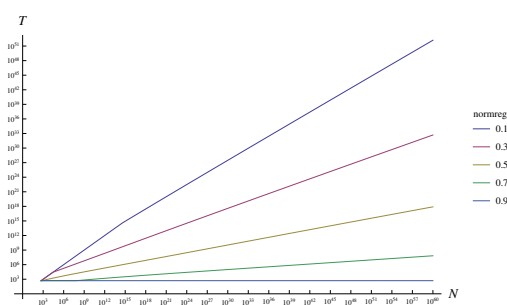

Figure 1: The upper bound on the regret from Corollary 1 as a function of the number of evaluations $T$. Each curve corresponds to a different domain size $N$ and is plotted for $T \in \{500, ..., N\}$, hence $N$ can be read from the end point of each curve.

Figure 2: An upper bound on the required number of evaluations $T$ as a function of the domain size $N$ (implied by Corollary 1). Each curve corresponds to a given regret which we want to achieve.

*This implies a bound on the ratio between the expected maximum found by the algorithm and the expected global maximum.*

*Proof.* The Gaussian prior is symmetric about $0$, as the mean is $0$ everywhere (i.e. the probability density satisfies $p(F = f) = p(F = -f) \; \forall f$). Since also both the EI2 and the UCB2 policies are symmetric, we have $\mathbb{E}\left[\hat{F}\right] = -\mathbb{E}\left[\check{F}\right]$ and hence

$$\mathbb{E}\left[\hat{\check{F}}\right] = \mathbb{E}\left[\hat{F}\right] - \mathbb{E}\left[\check{F}\right] = 2\mathbb{E}\left[\hat{F}\right] \tag{19}$$

and similarly $\mathbb{E}\left[\hat{\check{Y}}\right] = 2\mathbb{E}\left[\hat{Y}\right]$. Substituting this in Theorem 1, Corollary 1 follows. $\square$

In Figure 1, we plot this bound, and we observe that we obtain nontrivial regret bounds even when evaluating the function only at a small fraction of its domain. For instance, if we pick the curve that corresponds to $N = 10^{20}$ (the rightmost curve), and we choose $T = 10^6$, we achieve a regret of about $0.6$, as indicated in the figure. This means that we can expect to find a function value of about $40\%$ of the expected global maximum by evaluating the function at only a fraction of $10^{-14}$ of its domain, and this holds for any prior covariance $\Sigma$.

In Figure 2, we plot the upper bound on the required number of evaluations $T$, implied by Corollary 1, as a function of the domain size $N$. We observe that $T$ seems to scale polynomially with $N$ (i.e. linearly in log space) with an order that depends on the regret we want to achieve. Indeed, it is easy to see from Corollary 1 that $\forall \epsilon > 0 \; \exists K : \forall N \geq T \geq K :$

$$\text{normreg} \leq 1 - \frac{\sqrt{\log T}}{\sqrt{\log N}} + \epsilon \tag{20}$$

which implies that $\forall \epsilon > 0 \; \exists K : \forall N \geq T \geq K :$

$$T \leq N^{(1 - \text{normreg} + \epsilon)^2}. \tag{21}$$

This means for instance that, if we accept to achieve only $20\%$ of the global maximum, the required number of evaluations grows very slowly with about $T \approx N^{1/25}$, but still polynomially.

### 3.3.2 Lower Regret Bound for the Optimal Policy

The upper bound from Corollary 1 is tight, as we can see by comparing (20) to the following result:

**Lemma 1** (Lower Bound)**.** *For the instance of the problem defined in Section 2.1 with $\mu = \mathbf{0}$ and $\Sigma = \mathbf{I}$, the following lower bound on the regret holds for the optimal strategy: $\forall \epsilon > 0 \; \exists K : \forall N \geq T \geq K :$*

$$\text{normreg} \geq 1 - \frac{\sqrt{\log T}}{\sqrt{\log N}} - \epsilon. \tag{22}$$

*Proof.* Here, sampling without replacement is an optimal strategy, which yields the bound above, see Appendix I for the full proof. □

It follows from Lemma 1 that $\forall \epsilon > 0 \ \exists K : \forall N \geq T \geq K :$

$$T \geq N^{(1-\text{normreg}-\epsilon)^2}. \tag{23}$$

Hence, for a given regret, the required number of evaluations $T$ grows polynomially in the domain size $N$, even for the optimal policy, albeit with low degree. This means that if the domain size grows exponentially in the problem dimension, we will inherit this exponential growth also for the necessary number of evaluations. However, since our bounds are problem independent and hence worst-case in terms of the prior covariance $\Sigma$, this result does not exclude the possibility that for certain covariances we might be able to obtain polynomial scaling of the required number of evaluations in the dimension of the problem.

### 3.3.3 Lower Regret Bound for Prior-Independent Policies

It is important to note that a uniform random policy will not achieve the bound from Corollary 1 in general. In fact, despite the bound from Corollary 1 being independent of the prior $(\mu, \Sigma)$, it cannot be achieved by any policy that is independent of the prior:

**Lemma 2.** *For any optimization policy which does not depend on the prior $(\mu, \Sigma)$, there exists an instance of the problem defined in Section 2.1 where*

$$\text{normreg} \geq 1 - \frac{T}{N}, \tag{24}$$

*which is clearly worse for $T \ll N$ than the bound in Corollary 1.*

*Proof.* Suppose we construct a function that is zero everywhere except in one location. Since the policy has no knowledge of that location, it is possible to place it such that the policy will perform no better than random selection, which yields the regret above. See Appendix J for the full proof. □

## 3.4 Extension to Continuous Domains

For finite domains, we looked at the setting where the cardinality of the domain $N$ is much larger than the number of admissible evaluations $T$. The notion of domain size is less obvious in the continuous case, in the following we clarify this point before we discuss the results.

### 3.4.1 Problem Setting

In the continuous setting, we characterize the GP by $L_k$ and $\sigma$, which are properties of the kernel $k$:

$$|k(x, x) - k(x, y)| \leq L_k \|x - y\|_\infty \quad \forall x, y \in \mathcal{A} \tag{25}$$

$$k(x, x) \leq \sigma^2 \quad \forall x \in \mathcal{A} \tag{26}$$

where $\mathcal{A}$ is the $D$-dimensional unit cube (note that to use a domain other than the unit cube we can simply rescale). The setting we are interested in is

$$T \ll \left(\frac{L_k}{2\sigma^2}\right)^D =: m(L_k, \sigma, D). \tag{27}$$

As we discuss in more detail in Appendix C, $m(L_k, \sigma, D)$ corresponds to the number of points we would require to cover the domain such that we could acquire nonzero information about any point in the domain. Naturally, to guarantee that we can find the global optimum we would require at least that number of evaluations $T$. Here, in contrast, we consider the setting where $T$ is much smaller and only a small fraction of the GP can be explored.

### 3.4.2 Results

Here, we adapt Corollary 1 to continuous domains. The bound we propose in the following is based on a result from [Grünewälder et al., 2010], which states that for a centered Gaussian Process $(G_a)_{a \in \mathcal{A}}$ with domain $\mathcal{A}$ being the $D$-dimensional unit cube and a Lipschitz-continuous kernel $k$

$$|k(x,x) - k(x,y)| \leq L_k \|x - y\|_\infty \quad \forall x, y \in \mathcal{A}, \tag{28}$$

the regret of the optimal policy is bounded by

$$\mathbb{E}\left[\sup_{a \in \mathcal{A}} G(a) - \hat{Y}\right] \leq \sqrt{\frac{2L_k}{\lfloor T^{1/D} \rfloor}} \left(2\sqrt{\log(2T)} + 15\sqrt{D}\right). \tag{29}$$

Note that Grünewälder et al. [2010] state the bound for the more general case of Hölder-continuous functions, for simplicity of exposition we limit ourselves here to the case of Lipschitz-continuous kernels. Grünewälder et al. [2010] complement this bound with a matching lower bound (up to log factors). However, as we shall see, we can improve substantially on this bound in terms of its dependence on the Lipschitz constant $L_k$ if we assume that the variance is bounded, i.e., $k(x,x) \leq \sigma^2$. This is particularly relevant for GPs with short length scales (or, equivalently, large domains) and hence large $L_k$.

Interestingly, Grünewälder et al. [2010] obtain (29) using a policy that selects the actions a priori (by placing them on a grid), without any feedback from the observations made. Here, we will refine (29) by using this strategy for preselecting a large set of admissible actions offline and then selecting actions from this set using EI2 (Definition 2) or UCB2 (Definition 3) online. A reasoning along these lines yields the following bound:

**Theorem 2.** *For any centered Gaussian Process $(G_a)_{a \in \mathcal{A}}$, where $\mathcal{A}$ is the $D$-dimensional unit cube, with kernel $k$ such that*

$$|k(x,x) - k(x,y)| \leq L_k \|x - y\|_\infty \quad \forall x, y \in \mathcal{A} \tag{30}$$

$$\sqrt{k(x,x)} \leq \sigma \quad \forall x \in \mathcal{A} \tag{31}$$

*we obtain the following bound on the regret, if we follow the EI2 (Definition 2) or the UCB2 (Definition 3) strategy:*

$$\mathbb{E}\left[\sup_{a \in \mathcal{A}} G(a) - \hat{Y}\right] \leq \sqrt{\frac{2\log(L_k)}{T^{1/D}}} \left(2\sqrt{\log\left(2\left\lceil\frac{L_k}{\log(L_k)}T^{1/D}\right\rceil^D\right)} + 15\sqrt{D}\right) +$$

$$\sqrt{2}\sigma\left(\sqrt{D\log\left(\left\lceil\frac{L_k}{\log(L_k)}T^{1/D}\right\rceil\right)} - \left(1 - T^{-\frac{1}{2\sqrt{\pi}}}\right)\sqrt{\log\left(\frac{T}{3\log^{\frac{3}{2}}(T)}\right)}\right). \tag{32}$$

*This bound also holds when restricting EI2 or UCB2 to a uniform grid on the domain $\mathcal{A}$, where each side is divided into $\left\lceil\frac{L_k}{\log(L_k)}T^{1/D}\right\rceil$ segments. Finally, this bound converges to $0$ as $T \to \infty$.*

*Proof.* The idea here is to pre-select a set of $N$ points at locations $X_{1:N}$ on a grid and then sub-select points from this set during runtime using EI2 (Definition 2) or UCB2 (Definition 3). We bound the regret of this strategy by combining Theorem 1 with the main result from [Grünewälder et al., 2010], the full proof can be found in Appendix K. $\square$

The important point to note here is that in Theorem 2 the bound grows logarithmically in $L_k$, as opposed to the bound (29) from Grünewälder et al. [2010], which grows with $\sqrt{L_k}$. This means that for Gaussian Processes with high $L_k$, i.e. high variability, (32) is much lower than (29) (note that cuboid domains $\mathcal{A}$ can be rescaled to the unit cube by adapting the Lipschitz constant $L_k$ accordingly, hence a large domain is equivalent to a large Lipschitz constant). This allows for meaningful bounds even when the number of allowed evaluations is small relative to the domain-size of the function. We illustrate this in Figure 3. Consider for instance the case of $L_k = 10^5, \sigma = 1, D = 2, T = 10^5$, where the number of evaluations is far too low to explore the GP ($T = 10^5 \ll m(L_k, \sigma, D) = 2.5 \times 10^9$). We see from Figure 3a that our regret bounds (second-darkest cyan) remain low while the ones from Grünewälder et al. [2010] (second-darkest magenta) explode. Theorem 2 also provides another insight: To allow for straightforward optimization of the acquisition function (e.g. EI, UCB), the domain is often discretized in practical Bayesian optimization. Theorem 2 tells us how fine this discretization should be to still achieve performance guarantees.

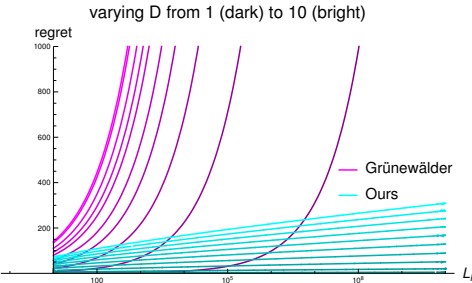

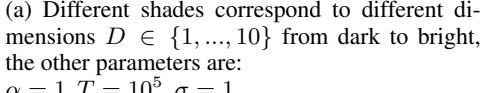

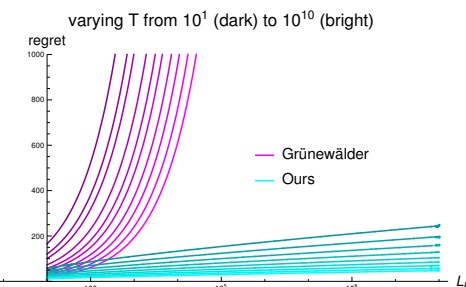

(a) Different shades correspond to different dimensions $D \in \{1, ..., 10\}$ from dark to bright, the other parameters are: $\alpha = 1, T = 10^5, \sigma = 1$.

(b) Different shades correspond to different numbers of evaluations $T \in \{10^1, 10^2, ..., 10^{10}\}$ from dark to bright, the other parameters are: $\alpha = 1, D = 5, \sigma = 1$.

Figure 3: The regret as a function of the Lipschitz constant $L_k$. Comparison of the bound from Grünewälder et al. [2010] (magenta) and ours (32) (cyan).

## 4   Relation to Standard EI and UCB

An empirical comparison (Appendix F) indicates EI2/UCB2 require more evaluations than EI/UCB to attain a given regret, but not more than twice as many. This matches our intuition: We would expect standard EI/UCB to perform better because i) any given step, evaluating at a potential maximizer instead of a minimizer will clearly lead to a larger immediate reduction in expected regret and ii) we would not expect an evaluation at a potential minimizer to provide any more useful information than an evaluation at a potential maximizer. Further, we would not expect EI2/UCB2 to perform much worse because a substantial fraction of its evaluations (in expectation half, for a centered GP) will be maximizations.

If we could prove that EI/UCB performs better than EI2/UCB2 for maximization, this would imply that the regret bounds presented above apply to EI/UCB. Unfortunately, proving this formally appears to be nontrivial. Nevertheless, we are able to give weaker regret bounds for standard EI/UCB which we discuss in the following.

### 4.1   Upper Bound on the Bayesian Simple Regret for Standard EI and UCB

Let us now consider the standard, one-sided versions of EI (Definition 6) and UCB (Definition 7), which are identical to EI2 (Definition 2) and UCB2 (Definition 3) except that we drop the second term in the max. We obtain the following version of Theorem 1:

**Theorem 3.** *For any instance $(N, T, \mu, \Sigma)$ of the problem defined in Section 2.1 with $N \geq T \geq 500$, if we follow either the **EI** (Definition 6) or the **UCB strategy** (Definition 7) we have*

$$\frac{\mathbb{E}\left[\hat{\check{F}}\right] - \mathbb{E}\left[\hat{Y} - \check{F}\right]}{\mathbb{E}\left[\hat{\check{F}}\right]} \leq 1 - \left(1 - T^{-\frac{1}{2\sqrt{\pi}}}\right)\sqrt{\frac{\log(T) - \log\left(3\log^{\frac{3}{2}}(T)\right)}{\log(N)}}. \tag{33}$$

*Proof.* The proof is very similar to the one of Theorem 1 and can be found in Appendix H. $\square$

Note that we marked the changes in bold. Now, instead of a guarantee on $\hat{\check{Y}}$, we provide a guarantee on $\hat{Y} - \check{F}$, i.e. the difference between the best obtained value and the function minimum. We can then derive from that a version of Corollary 1:

**Corollary 2.** *For any instance $(N, T, \mu, \Sigma)$ of the problem defined in Section 2.1 with zero mean $\mu_n = 0 \ \forall n$ and $N \geq T \geq 500$, if we follow either **standard EI** (Definition 6) or **UCB strategy** (Definition 7), we have*

$$normreg \leq \mathbf{2}\left(1 - \left(1 - T^{-\frac{1}{2\sqrt{\pi}}}\right)\sqrt{\frac{\log(T) - \log\left(3\log^{\frac{3}{2}}(T)\right)}{\log(N)}}\right). \tag{34}$$

*Proof.* The proof is analogous to the one of Corollary 1.

$\square$

The important thing to note here is the appearance of the factor 2 in the bound. This means that asymptotically we have

$$\mathrm{normreg} \leq \mathbf{2} \left( 1 - \frac{\sqrt{\log T}}{\sqrt{\log N}} \right) + \epsilon \tag{35}$$

and

$$T \leq N^{(1 - \mathrm{normreg}/\mathbf{2} + \epsilon)} \tag{36}$$

which is weaker compared to (20) and (21). We believe that this gap is not due to EI/UCB actually performing worse than EI2/UCB2, but rather an artifact of the proof.

## 5 Limitations

While we believe that the results above are insightful, there are a number of limitations one should be aware of:

As discussed in Section 4, the regret bounds we derive for standard EI and UCB are weaker than the ones for EI2 and UCB2, despite the intuition and empirical evidence that EI/UCB most likely perform no worse for maximization than EI2/UCB2. It would be interesting to close this gap.

Another limitation is that the bounds only hold for the noise-free setting. We believe that this limitation is acceptable because the problem of noisy observations is mostly orthogonal to the problem studied herein. Furthermore, the naive solution of reducing the noise by evaluating multiple times at each point leads to qualitatively similar regret bounds, see Appendix D for a more detailed discussion.

Further, the bounds from Corollary 1, Theorem 2, and Corollary 2 only hold for zero prior mean (equivalently, constant prior mean). This assumption is not uncommon in the GP optimization literature (see e.g. Srinivas et al. [2010]) but it may be limiting if one has prior knowledge about where good function values lie. It is likely possible to extend the results in this article to arbitrary prior means.

Finally, there are limitations that hold generally for the GP optimization literature: 1) In a naive implementation, the computational cost is cubic in the number of evaluations $T$ and 2) the assumption that the true function is drawn from a Gaussian Process is typically not realistic and only made for analytical convenience. It is hence not clear whether the relations we uncovered herein apply to realistic optimization settings or if they are mostly an artifact of the GP assumption.

Summarizing, it is clear that the results in the present paper have little direct practical relevance. Instead, the intention is to develop a theoretical understanding of the problem setting.

## 6 Conclusion

We have characterized GP optimization in the setting where finding the global optimum is impossible because the number of evaluations is too small with respect to the domain size. We derived regret-bounds for the finite-arm setting which are independent of the prior covariance, and we showed that they are tight. Further, we derived regret-bounds for GP optimization in continuous domains that depend on the Lipschitz constant of the covariance function and the maximum variance. In contrast to previous work, our bounds are non-vacuous even when the domain size is very large relative to the number of evaluations. Therefore, they provide novel insights into the performance of GP optimization in this challenging setting. In particular, they show that even when the number of evaluations is far too small to find the global optimum, we can find nontrivial function values (e.g. values that achieve a certain ratio with the optimal value).

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
