## A  Relation to Adaptive Submodularity

Submodularity is a property of set functions with far-reaching implications. Most importantly here, it allows for efficient approximate optimization Nemhauser et al. [1978], given the additional condition of monotonicity. This fact has been exploited in many information gathering applications, see Krause and Golovin [2014] for an overview.

In Golovin and Krause [2011], the authors extend the notion of submodularity to adaptive problems, where decisions are based on information acquired online. This is precisely the setting we consider in the present paper. However, as we will show shortly, our function maximization problem is not submodular. Nevertheless, our proof is inspired by the notion of adaptive sumodularity.

Consider the following definitions, which we adapted to the notation in the present paper:

**Definition 4.** *(Golovin and Krause [2011]) The Conditional Expected Marginal Benefit with respect to some utility function $u$ is defined as*

$$\Delta_u(a|a_{1:t}, y_{1:t}) = \mathbb{E}\left[u(F, A_{1:t+1}) - u(F, A_{1:t})|Y_{1:t} = y_{1:t}, A_{1:t} = a_{1:t}, A_{t+1} = a\right]. \quad (37)$$

**Definition 5.** *(Golovin and Krause [2011]) Adaptive submodularity holds if for any $t \leq k \in \mathbb{N}$, any $a_{1:k}, y_{1:k}$ and any $a$ we have*

$$\Delta_u(a|a_{1:t}, y_{1:t}) \geq \Delta_u(a|a_{1:k}, y_{1:k}). \quad (38)$$

Intuitively, in an adaptively submodular problem the expected benefit of any given action $a$ decreases the more information we gather. Golovin and Krause [2011] show that if a problem is adaptively submodular (along with some other condition), then the greedy policy will converge exponentially to the optimal policy.

### A.1  Gaussian-Process Optimization is not Adaptively Submodular

In the following we make a simple argument why GP optimization is not generally adaptively submodular. It is not entirely clear what is the right utility function $u$, but our argument holds for any plausible choice.

Consider a function $F_{1:N}$ with all values mutually independent, except for $F_1$ and $F_2$ which are negatively correlated. Further, suppose that we made an observation $y_1$ which is far larger than the upper confidence bounds on $F_1$ and $F_2$. Any reasonable choice of utility function would yield an extremely small conditional expected marginal benefit for $A_2 = 2$, since we would not expect this to give us any information about the optimum. Now suppose we evaluate the function at $A_2 = 1$ and observe a $y_2$ such that the posterior mean of $F_2$ is approximately equal to $y_1$. Now, the conditional expected marginal benefit of evaluating at $A_3 = 2$ should be substantial for any reasonable utility, since the maximum might lie at that point. More generally, through unlikely observations the GP landscape can change completely and points which seemed uninteresting before can become interesting, which violates the diminishing-returns property of adaptive submodularity Definition 5.

## B  Relation to GP-UCB

The bounds from Srinivas et al. [2010] are only meaningful if we are allowed to make a sufficient number of evaluations to attain low uncertainty over the entire GP. The reason is that these bounds depend on a term called the information gain $\gamma_T$, which represents the maximum information that can be acquired about the GP with $T$ evaluations. As long as the GP still has large uncertainty in some areas, each additional evaluation may add a substantial amount of information (there is no saturation) and $\gamma_T$, and hence the cumulative regret, will keep growing.

To see this, consider Lemma 5.3 in Srinivas et al. [2010] (we use a slightly different notation here): The information gain of a set of points $X = x_1, ..., x_T$ can be expressed as

$$G(X) := I(y_X; f_X) = \frac{1}{2} \sum_{t=1}^{T} \log(1 + \sigma_y^{-2}\sigma^2(x_t|x_{1:t-1})) \quad (39)$$

where $\sigma^2(x_t|x_{1:t-1})$ is the predictive variance after evaluating at $x_{1:t-1}$ and $\sigma_y^2$ is the variance of the observation noise[2]. Hence, we can write the information gain for $T + 1$ points as

$$G(X \cup \{x_{T+1}\}) = G(X) + \frac{1}{2}\log(1 + \sigma_y^{-2}\sigma^2(x_{T+1}|x_{1:T})). \tag{40}$$

Now let $X^* := \max_{X:|X|=T} G(X)$ be the points that maximize the information gain. By definition (see equation 7 in Srinivas et al. [2010]), we have

$$\gamma_T := G(X^*) \tag{41}$$

that is, $\gamma_T$ is the maximum information that can be acquired using $T$ points. For $T + 1$ points we have

$$\gamma_{T+1} = \max_{X, x_{T+1}} G(X \cup \{x_{T+1}\}) \tag{42}$$

$$\geq \max_{x_{T+1}} G(X^* \cup \{x_{T+1}\}) \tag{43}$$

where the inequality follows from the fact that maximizing over $X, x_{T+1}$ jointly will at least yield as high a value as just picking $X^*$ from the previous optimization and optimizing only over $x_{T+1}$. Plugging in (40), we have

$$\gamma_{T+1} \geq G(X^*) + \frac{1}{2}\max_{x_{T+1}}\log(1 + \sigma_y^{-2}\sigma^2(x_{T+1}|x_{1:T}^*)) \tag{44}$$

and hence

$$\gamma_{T+1} \geq \gamma_T + \frac{1}{2}\max_{x_{T+1}}\log(1 + \sigma_y^{-2}\sigma^2(x_{T+1}|x_{1:T}^*)). \tag{45}$$

This means that if $T$ is not large enough to explore the GP reasonably well everywhere (i.e., there are still $x$ such that $\sigma^2(x|x_{1:T}^*)$ is large), then adding an observation can add substantial information, i.e. $\gamma_{T+1}$ is substantially larger than $\gamma_T$ (which means the regret grows substantially).

As a more concrete case, suppose we have a GP which a priori has a uniform variance $\sigma^2(x) = s^2$ $\forall x$. In addition, suppose that the GP domain is large with respect to $T$, in the sense that it is not possible to reduce the variance everywhere substantially by observing $T$ (or less) points, i.e. we have $\max_{x_t} \sigma(x_t|x_{1:t-1}) \approx s \ \forall x_{1:t-1}, t \leq T+1$. We hence have

$$\gamma_T = \max_{x_{1:T}} \frac{1}{2}\sum_{t=1}^{T}\log(1 + \sigma_y^{-2}\sigma^2(x_t|x_{1:t-1})) \tag{46}$$

$$\approx \frac{1}{2}T\log(1 + \sigma_y^{-2}s^2). \tag{47}$$

This linear growth will continue until $T$ is large enough such that the uncertainty of the GP can be reduced substantially everywhere.

Since the bound on the cumulative regret $R_T$ is of the form $\sqrt{T\gamma_T}$ (see Theorem 1 in Srinivas et al. [2010]) it will hence also grow linearly in $T$. Srinivas et al. [2010] then bound the suboptimality of the optimization by the average regret $R_T/T$ (see the paragraph on regret in Section 2 of Srinivas et al. [2010]), which does not decrease as long as $R_T$ grows linearly in $T$.

## C  The Continuous-Domain Setting

In the continuous setting, we characterize the GP by $L_k$ and $\sigma$, which are properties of the kernel $k$:

$$|k(x, x) - k(x, y)| \leq L_k \|x - y\|_\infty \quad \forall x, y \in \mathcal{A} \tag{48}$$

$$k(x, x) \leq \sigma^2 \quad \forall x \in \mathcal{A} \tag{49}$$

---

[2]Note that the information gain goes to infinity as the observation noise $\sigma_y$ goes to zero, which is in fact another reason why the results from Srinivas et al. [2010] are not directly applicable to our setting. However, this is a technicality that can be resolved (in the most naive way, one could add artificial noise).

where $\mathcal{A}$ is the $D$-dimensional unit cube (note that to use a domain other than the unit cube we can simply rescale). The setting we are interested in is

$$T \ll \left(\frac{L_k}{2\sigma^2}\right)^D =: m(L_k, \sigma, D), \tag{50}$$

which implies that a large part of the GP may remain unexplored, as will become clear in the following comparison to related work:

As discussed in the introduction and in Appendix B, the results from Srinivas et al. [2010] only apply when we can reduce the maximum variance of the GP using $T$ evaluations. This would require that we can acquire information on each point $x$ that has maximum prior variance $k(x, x) = \max_z k(z, z) = \sigma^2$. In order to ensure that we gather nonzero information on such a point $x$, we have to make sure to evaluate at least one point $y$ such that $k(x, y) > 0$ (or, more realistically, $k(x, y) > \epsilon$, which would lead to a qualitatively similar result), which is equivalent to the condition

$$|k(x, x) - k(x, y)| < k(x, x) = \sigma^2, \tag{51}$$

which we can ensure by

$$L_k \|x - y\|_\infty < \sigma^2 \tag{52}$$

or equivalently by

$$\|x - y\|_\infty < \frac{\sigma^2}{L_k}. \tag{53}$$

This statement says that $x$ has to be within a cube centered at $y$ with sidelength $2\sigma^2/L_k$. To ensure that this holds for all $x \in \mathcal{A}$ (since in the worst case they all have prior variance $\sigma^2$, which is typical), we need to cover the domain with

$$T > \left(\frac{L_k}{2\sigma^2}\right)^D = m(L_k, \sigma, D) \tag{54}$$

cubes and hence evaluations.

## D  A Note on Observation Noise

Our goal here was to focus on the issue of large domains, without the added difficulty of noisy observations, such as to allow a clearer view of the core problem. Interestingly, the proofs apply practically without any changes to the setting with observation noise. The caveat is that the regret bounds are on the largest **noisy observation** $\hat{Y}$ rather than the largest retrieved **function value** $\max_t F_{A_t}$ (the two are identical in the noise-free setting).

As a naive way of obtaining regret bounds on $\max_t F_{A_t}$, one could simply evaluate each point $n$ times and use the average observation as a pseudo observation. Choosing $n$ large enough, all pseudo observations $Y_{1:T}$ will be close to their respective function values $F_{A_{1:T}}$ with high probability. To guarantee that all $T$ pseudo observations are within $\epsilon$ of the true function values with probability $\delta$, we would need $n = \log(T/\delta)f(\sigma_y, \epsilon)$ (this follows from union bound over $T$ observations), where $f$ is some function that is not relevant here and $\sigma_y$ is the noise standard deviation. We can now simply replace $T$ with $T/(\log(T/\delta)f(\sigma_y, \epsilon))$ in all the theorems (to be precise, we would also have to add $\epsilon$ to the regret, but it can be made arbitrarily small). While this solution is impractical, it is interesting to note that the dependence of the resulting regret-bounds on the domain size $N$ and Lipschitz constant $L_k$ does not change. The dependence on $T$ is also identical, up to a $\log$ factor. This suggests that the relations we uncovered in this paper between the regret, the number of evaluations $T$, the domain size $N$, the Lipschitz constant $L_k$ remain qualitatively the same in the presence of observation noise.

## E  Definitions of Standard EI/UCB

**Definition 6** (EI)**.** *An agent follows the EI strategy when it picks its actions according to*

$$A_{t+1} = \underset{n \in [N]}{\operatorname{argmax}} \operatorname{ei}\left(\hat{Y}_t | M_t^n, \sqrt{C_t^{nn}}\right) \tag{55}$$

*with the expected improvement* ei *as defined in Definition 1.*

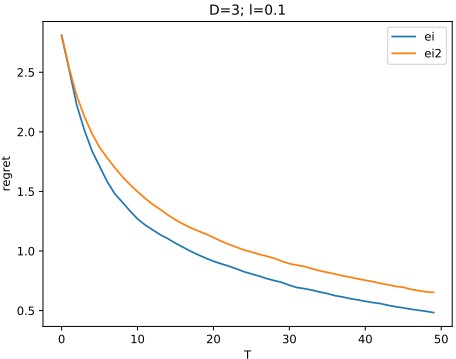

Figure 4: The empirical expected regret as a function of the number of evaluations $T$.

**Definition 7** (UCB). *An agent follows the UCB strategy when it picks its actions according to*

$$A_{t+1} = \operatorname*{argmax}_{n\in[N]} \left( M_t^n + \sqrt{C_t^{nn} 2\log N} \right). \tag{56}$$

## F Empirical Comparison between EI/UCB and EI2/UCB2

We conducted a number of within-model experiments, where the ground-truth function is a sample drawn from the GP. The continuous-domain experiments use the GPy [since 2012] library and the discrete-domain experiments use scikit-learn (Pedregosa et al. [2011]). The code for all the experiments is publicly accessible[3].

### F.1 Continuous Domain

We defined a GP $G$ on a $D$-dimensional unit cube with a squared-exponential kernel with length scale $l$. The smaller the length-scale and the larger the dimensionality, the harder the problem. For the experiments that follow, we chose the ranges of $D, l$ such that we cover the classical setting (where the global optimum can be identified) as well as the large-domain setting (where this is not possible). To make this scenario computationally tractable, we discretize the domain using a grid with 1000 points, and we allow the algorithm to evaluate at $T = 50$ points.

The true expected regret

$$\mathbb{E}\left[\sup_{a\in\mathcal{A}} G(a) - \hat{Y}\right] = r(l, D, T) \tag{57}$$

is a function of the length-scale $l$, the dimension $D$, and the number of function evaluations $T$. We compute this quantity empirically using 1000 samples (i.e. 1000 randomly-drawn ground-truth functions). In Figure 4, for instance, we plot this empirical expected regret as a function of the number of evaluations $T$. In this example, EI performs slightly better than EI2. To gain a more quantitative understanding, it is instructive to look at how many evaluations $T$ are required to attain a given regret $R = r(l, D, T)$:

$$T = t(R, l, D). \tag{58}$$

We can then compare the required number of steps for EI and EI2:

$$\frac{t_{ei}(R, l, D)}{t_{ei2}(R, l, D)}, \tag{59}$$

which we report in Table 1. In one entry EI2 appears to perform slightly better, we have $t_{ei2} = 1.06 t_{ei}$. This is may be due to the variance of the empirical estimation. In all other entries, we have $0.5 t_{ei2} \leq t_{ei} \leq t_{ei2}$, which means that EI always reaches the given expected regret $R$ faster than EI2, but not more than twice as fast. This is what we intuitively expected: EI should do

---

[3] https://github.com/mwuethri/regret-Bounds-for-Gaussian-Process-Optimization-in-Large-Domains

Table 1: Fraction $\frac{t_{ei}(R,l,D)}{t_{ei2}(R,l,D)}$ for continuous domains. NaN entries correspond to the case where the given regret was not attained after $T = 50$ evaluations.

| D | l | 2.5 | 2.0 | 1.5 | 1.0 | 0.5 |
|---|---|-----|-----|-----|-----|-----|
| 1 | 0.003 | 1.00 | 1.00 | 0.87 | 0.84 | NaN |
|   | 0.010 | 1.00 | 1.00 | 0.80 | 0.80 | 0.74 |
|   | 0.030 | 1.00 | 1.00 | 1.00 | 1.00 | 0.78 |
|   | 0.100 | 1.00 | 1.00 | 1.00 | 1.00 | 0.75 |
|   | 0.300 | 1.00 | 1.00 | 1.00 | 1.00 | 1.00 |
| 2 | 0.003 | 1.00 | 1.00 | 1.00 | 1.00 | NaN |
|   | 0.010 | 1.00 | 0.86 | 1.00 | NaN | NaN |
|   | 0.030 | 0.67 | 1.00 | 0.91 | 0.71 | NaN |
|   | 0.100 | 1.00 | 1.00 | 0.80 | 0.78 | 0.75 |
|   | 0.300 | 1.00 | 1.00 | 1.00 | 1.00 | 0.83 |
| 3 | 0.003 | 1.00 | 1.00 | 1.00 | 1.00 | NaN |
|   | 0.010 | 1.00 | 1.00 | 1.00 | NaN | NaN |
|   | 0.030 | 1.00 | 1.00 | 1.06 | NaN | NaN |
|   | 0.100 | 1.00 | 1.00 | 0.73 | 0.69 | NaN |
|   | 0.300 | 1.00 | 1.00 | 0.75 | 0.86 | 0.73 |
| 4 | 0.003 | 1.00 | 1.00 | 1.00 | NaN | NaN |
|   | 0.010 | 1.00 | 1.00 | 1.00 | NaN | NaN |
|   | 0.030 | 1.00 | 1.00 | 1.00 | NaN | NaN |
|   | 0.100 | 1.00 | 1.00 | 0.84 | NaN | NaN |
|   | 0.300 | 1.00 | 0.75 | 0.86 | 0.71 | 0.70 |

better than EI2, because it does not waste evaluations on minimization, but not much better, since in expectation every second evaluation of EI2 is a maximization. Note that the entries which are 1, i.e. both algorithms perform equally well, correspond to i) particularly simple settings (large $l$, low $D$) where both algorithms find good values in just a handful of evaluations or ii) particularly hard settings where there is no essentially no correlation between different points in the discretized domain.

For UCB and UCB2 (where we used a fixed confidence level) we obtain similar results, see Table 2.

## F.2 Band Covariance Matrices

Next, we compare EI/UCB with EI2/UCB2 in the discrete setting with $N = 100$. We use band covariance matrices, where the diagonal elements are equal to $1$ and there are a number of nonzero elements to the right and the left of the diagonal. We vary width of this band and the value the off-diagonal elements take, we report the results in Table 3 for EI vs EI2 and in Table 4 for UCB vs UCB2. Similarly to the case of continuous domains, we see that $0.5t_{ei2} \leq t_{ei} \leq t_{ei2}$ (and the equivalent for UCB).

## F.3 Randomly-Sampled Covariance Matrices

Finally, we sample covariances (of size $N = 200$) randomly from an inverse Wishart distribution (with $400$ degrees of freedom and identity scale matrix). We report the results in Table 5 for EI vs EI2 and in Table 6 for UCB vs UCB2. As in the previous experiments, we see that $0.5t_{ei2} \leq t_{ei} \leq t_{ei2}$ (and the equivalent for UCB).

Table 2: Fraction $\frac{t_{ucb}(R,l,D)}{t_{ucb2}(R,l,D)}$ for continuous domains. NaN entries correspond to the case where the given regret was not attained after $T = 50$ evaluations.

| D | l | 2.5 | 2.0 | 1.5 | 1.0 | 0.5 |
|---|---|---|---|---|---|---|
| 1 | 0.003 | 1.00 | 1.00 | 0.89 | 0.86 | NaN |
|   | 0.010 | 1.00 | 1.00 | 0.80 | 0.90 | 0.81 |
|   | 0.030 | 1.00 | 1.00 | 1.00 | 1.00 | 0.90 |
|   | 0.100 | 1.00 | 1.00 | 1.00 | 1.00 | 1.00 |
|   | 0.300 | 1.00 | 1.00 | 1.00 | 1.00 | 1.00 |
| 2 | 0.003 | 1.00 | 1.00 | 1.00 | 1.00 | NaN |
|   | 0.010 | 1.00 | 0.86 | 1.00 | NaN | NaN |
|   | 0.030 | 0.67 | 0.83 | 0.83 | 0.77 | NaN |
|   | 0.100 | 1.00 | 1.00 | 0.80 | 0.73 | 0.78 |
|   | 0.300 | 1.00 | 1.00 | 1.00 | 1.00 | 0.83 |
| 3 | 0.003 | 1.00 | 1.00 | 1.00 | 1.00 | NaN |
|   | 0.010 | 1.00 | 1.00 | 1.00 | NaN | NaN |
|   | 0.030 | 1.00 | 1.00 | 1.00 | NaN | NaN |
|   | 0.100 | 1.00 | 0.83 | 0.75 | 0.76 | NaN |
|   | 0.300 | 1.00 | 1.00 | 0.75 | 1.00 | 0.71 |
| 4 | 0.003 | 1.00 | 1.00 | 1.00 | NaN | NaN |
|   | 0.010 | 1.00 | 1.00 | 1.00 | NaN | NaN |
|   | 0.030 | 1.00 | 1.00 | 1.00 | NaN | NaN |
|   | 0.100 | 1.00 | 0.86 | 0.83 | NaN | NaN |
|   | 0.300 | 1.00 | 0.75 | 0.87 | 0.75 | 0.66 |

Table 3: Fraction $\frac{t_{ei}(\text{band\_size, band\_corr})}{t_{ei2}(\text{band\_size, band\_corr})}$ for finite band covariance matrices. The covariance matries are identity matrices with band_size many elements with value band_corr added to each side of the diagonal. NaN entries correspond to the case where the given regret was not attained after $T = 20$ evaluations.

| band_size | band_corr | 2.00 | 1.55 | 1.10 | 0.65 | 0.20 |
|---|---|---|---|---|---|---|
| 0 | 0.00 | 1.0 | 1.00 | 1.00 | NaN | NaN |
| 2 | -0.20 | 1.0 | 0.75 | 0.86 | 0.82 | NaN |
| 3 | 0.20 | 1.0 | 1.00 | 0.87 | 0.89 | NaN |
| 5 | -0.10 | 1.0 | 1.00 | 1.00 | 0.94 | NaN |
|   | 0.20 | 1.0 | 1.00 | 1.00 | 0.88 | NaN |
| 10 | 0.10 | 1.0 | 1.00 | 1.00 | 0.94 | NaN |
| 40 | 0.05 | 1.0 | 1.00 | 0.87 | 0.94 | NaN |

Table 4: Fraction $\frac{t_{ucb}(\text{band\_size, band\_corr})}{t_{ucb2}(\text{band\_size, band\_corr})}$ for finite band covariance matrices. The covariance matries are identity matrices with band_size many elements with value band_corr added to each side of the diagonal. NaN entries correspond to the case where the given regret was not attained after $T = 20$ evaluations.

| band_size | band_corr | 2.00 | 1.55 | 1.10 | 0.65 | 0.20 |
|---|---|---|---|---|---|---|
| 0 | 0.00 | 1.0 | 1.0 | 1.00 | NaN | NaN |
| 2 | -0.20 | 1.0 | 1.0 | 0.86 | 0.87 | NaN |
| 3 | 0.20 | 1.0 | 1.0 | 0.87 | 0.89 | NaN |
| 5 | -0.10 | 1.0 | 1.0 | 1.00 | 0.88 | NaN |
|   | 0.20 | 1.0 | 1.0 | 1.00 | 0.88 | NaN |
| 10 | 0.10 | 1.0 | 1.0 | 1.00 | 0.83 | NaN |
| 40 | 0.05 | 1.0 | 1.0 | 1.00 | 1.00 | NaN |

Table 5: Fraction $\frac{t_{ei}(\text{wishart\_seed})}{t_{ei2}(\text{wishart\_seed})}$ for covariance matrices drawn from a Wishart distribution. NaN entries correspond to the case where the given regret was not attained after $T = 30$ evaluations.

| | 0.20 | 0.15 | 0.10 | 0.05 | 0.00 |
|---|---|---|---|---|---|
| wishart_seed | | | | | |
| 1 | 1.0 | 0.67 | 0.83 | 0.78 | NaN |
| 2 | 1.0 | 1.00 | 0.83 | 0.87 | NaN |
| 3 | 1.0 | 1.00 | 0.83 | 0.83 | NaN |
| 4 | 1.0 | 1.00 | 1.00 | 0.82 | NaN |
| 5 | 1.0 | 1.00 | 1.00 | 0.82 | NaN |

Table 6: Fraction $\frac{t_{ucb}(\text{wishart\_seed})}{t_{ucb2}(\text{wishart\_seed})}$ for covariance matrices drawn from a Wishart distribution. NaN entries correspond to the case where the given regret was not attained after $T = 30$ evaluations.

| | 0.20 | 0.15 | 0.10 | 0.05 | 0.00 |
|---|---|---|---|---|---|
| wishart_seed | | | | | |
| 1 | 1.0 | 1.0 | 0.83 | 0.82 | NaN |
| 2 | 1.0 | 1.0 | 0.83 | 0.87 | NaN |
| 3 | 1.0 | 1.0 | 0.83 | 0.88 | NaN |
| 4 | 1.0 | 1.0 | 1.00 | 0.87 | NaN |
| 5 | 1.0 | 1.0 | 1.00 | 0.81 | NaN |

# G    Proof of Theorem 1

In this section we prove Theorem 1. As we have seen in the previous section, our problem is not adaptively submodular. Nevertheless, the following proof is heavily inspired by the proof in Golovin and Krause [2011]. We derive a less strict condition than adaptive submodularity which is applicable to our problem and implies that we converge exponentially to the optimum·$\beta$:

**Lemma 3.** *For any problem of the type defined in Section 2.1, we have for any $\alpha, \beta > 0$*

$$\beta \mathbb{E}\left[\hat{\tilde{F}}\right] - \mathbb{E}\left[\hat{\tilde{Y}}_t\right] \leq \alpha \left(\mathbb{E}\left[\hat{\tilde{Y}}_{t+1}\right] - \mathbb{E}\left[\hat{\tilde{Y}}_t\right]\right) \quad \forall t \in \{1 : T-1\} \tag{60}$$

$$\Downarrow$$

$$(1 - e^{-\frac{T-1}{\alpha}})\beta \mathbb{E}\left[\hat{\tilde{F}}\right] \leq \mathbb{E}\left[\hat{\tilde{Y}}_T\right]. \tag{61}$$

*i.e. the first inequality implies the second inequality.*

*Proof.* The proof is closely related to the adaptive submodularity proof by Golovin and Krause [2011]. Defining $\delta_t := \beta \mathbb{E}\left[\hat{\tilde{F}}\right] - \mathbb{E}\left[\hat{\tilde{Y}}_t\right] \forall t \in \{1 : T\}$, we can rewrite the first inequality as

$$\delta_t \leq \alpha(\delta_t - \delta_{t+1}) \quad \forall t \in \{1 : T-1\}$$

$$\delta_{t+1} \leq (1 - \frac{1}{\alpha})\delta_t \quad \forall t \in \{1 : T-1\}$$

Since the function $e^x$ is convex, we have $e^x \geq 1 + x \quad \forall x$. Using this fact, we obtain the inequality

$$\delta_{t+1} \leq e^{-\frac{1}{\alpha}}\delta_t \quad \forall t \in \{1 : T-1\}$$

$$\delta_T \leq e^{-\frac{T-1}{\alpha}}\delta_1$$

Now we can substitute $\delta_T = \beta \mathbb{E}\left[\hat{\tilde{F}}\right] - \mathbb{E}\left[\hat{\tilde{Y}}_T\right]$ and $\delta_1 = \beta \mathbb{E}\left[\hat{\tilde{F}}\right] - \mathbb{E}\left[\hat{\tilde{Y}}_1\right] = \beta \mathbb{E}\left[\hat{\tilde{F}}\right]$ (since $\hat{\tilde{Y}}_1 = 0$):

$$\beta\mathbb{E}\left[\hat{\mathring{F}}\right] - \mathbb{E}\left[\mathring{\hat{Y}}_T\right] \le e^{-\frac{T-1}{\alpha}}\beta\mathbb{E}\left[\hat{\mathring{F}}\right]$$

$$(1 - e^{-\frac{T-1}{\alpha}})\beta\mathbb{E}\left[\hat{\mathring{F}}\right] \le \mathbb{E}\left[\mathring{\hat{Y}}_T\right].$$

$\square$

Hence, if for some $\alpha, \beta > 0$ we can show that (60) holds, Lemma 3 yields a lower bound on the expected utility.

## G.1 Specialization for the Extremization Problem

**Lemma 4.** *For any problem of the type defined in Section 2.1 we have for any $\alpha, \beta > 0$*

$$\mathbb{E}\left[\beta\hat{\mathring{F}} - \mathring{\hat{Y}}_t | m_t, c_t, \hat{y}_t, \check{y}_t\right] \le \alpha\mathbb{E}\left[\mathring{\hat{Y}}_{t+1} - \mathring{\hat{Y}}_t | m_t, c_t, \hat{y}_t, \check{y}_t\right] \tag{62}$$
$$\forall t \in \{1 : T-1\}, m_t \in \mathbb{R}^N, c_t \in \mathbb{S}_+^N, \hat{y}_t \ge \check{y}_t \in \mathbb{R}$$

$$\Downarrow$$

$$(1 - e^{-\frac{T-1}{\alpha}})\beta\mathbb{E}\left[\hat{\mathring{F}}\right] \le \mathbb{E}\left[\mathring{\hat{Y}}\right]. \tag{63}$$

*Proof.* It is easy to see that the implication

$$\mathbb{E}\left[\beta\hat{\mathring{F}} - \mathring{\hat{Y}}_t | m_t, c_t, \hat{y}_t, \check{y}_t\right] \le \alpha\mathbb{E}\left[\mathring{\hat{Y}}_{t+1} - \mathring{\hat{Y}}_t | m_t, c_t, \hat{y}_t, \check{y}_t\right] \tag{64}$$
$$\forall t \in [T-1], m_t \in \mathbb{R}^N, c_t \in \mathbb{S}_+^N, \hat{y}_t \ge \check{y}_t \in \mathbb{R}$$

$$\Downarrow$$

$$\mathbb{E}\left[\beta\hat{\mathring{F}} - \mathring{\hat{Y}}_t\right] \le \alpha\mathbb{E}\left[\mathring{\hat{Y}}_{t+1} - \mathring{\hat{Y}}_t\right] \quad \forall t \in \{1 : T-1\}. \tag{65}$$

holds, since taking the expectation with respect to $M_t, C_t, \hat{Y}_t, \check{Y}_t$ on both sides of the first line yields the second line. Since (65) is identical to Lemma 3, the desired implication follows from these two implications. $\square$

To prove that (62) holds, we will derive a lower bound for the right-hand side and an upper bound for the left-hand side.

## G.2 Lower Bound for the Right-Hand Side

**Lemma 5.** *For any instance $(N, T, \mu, \Sigma)$ of the problem defined in Section 2.1, if we follow either the EI2 (Definition 2) or the UCB2 (Definition 3) strategy, we have*

$$\mathbb{E}\left[\mathring{\hat{Y}}_{t+1} - \mathring{\hat{Y}}_t | m_t, c_t, \hat{y}_t, \check{y}_t\right] \tag{66}$$
$$\ge \max\left\{\text{ei}\left(\hat{y}_t \middle| m_t^{n_{ucb}}, \sqrt{c_t^{n_{ucb}n_{ucb}}}\right), \text{ei}\left(-\check{y}_t \middle| -m_t^{n_{ucb}}, \sqrt{c_t^{n_{ucb}n_{ucb}}}\right)\right\}$$

*with* ei *as defined in Definition 1 and*

$$n_{ucb} := \underset{n \in [N]}{\arg\max} \max\left\{-\hat{y}_t + m_t^n + \sqrt{c_t^{nn}2\log N}, \check{y}_t - m_t^n + \sqrt{c_t^{nn}2\log N}\right\} \tag{67}$$

*for any $t \in \{1 : T-1\}, m_t \in \mathbb{R}^N, c_t \in \mathbb{S}_+^N, \hat{y}_t \ge \check{y}_t \in \mathbb{R}$.*

*Proof.* Developing the expectation on the left hand side of (62) we have

$$\mathbb{E}\left[\hat{\breve{Y}}_{t+1} - \hat{\breve{Y}}_t \middle| m_t, c_t, \hat{y}_t, \breve{y}_t\right] \tag{68}$$

$$= \mathbb{E}\left[\max\left\{F_{A_{t+1}} - \hat{y}_t, 0\right\} + \max\left\{-F_{A_{t+1}} + \breve{y}_t, 0\right\} \middle| m_t, c_t, \hat{y}_t, \breve{y}_t\right] \tag{69}$$

$$= \mathbb{E}\left[\text{ei}\left(\hat{y}_t \middle| m_t^{A_{t+1}}, \sqrt{c_t^{A_{t+1}A_{t+1}}}\right) + \text{ei}\left(-\breve{y}_t \middle| -m_t^{A_{t+1}}, \sqrt{c_t^{A_{t+1}A_{t+1}}}\right) \middle| m_t, c_t, \hat{y}_t, \breve{y}_t\right] \tag{70}$$

$$\geq \mathbb{E}\left[\max\left\{\text{ei}\left(\hat{y}_t \middle| m_t^{A_{t+1}}, \sqrt{c_t^{A_{t+1}A_{t+1}}}\right), \text{ei}\left(-\breve{y}_t \middle| -m_t^{A_{t+1}}, \sqrt{c_t^{A_{t+1}A_{t+1}}}\right)\right\} \middle| m_t, c_t, \hat{y}_t, \breve{y}_t\right] \tag{71}$$

where we have used Definition 1, and the inequality follows from the fact that the expected improvement (ei) is always $\geq 0$. The action $A_{t+1}$ is a function of $M_t, C_t, \hat{Y}_t, \breve{Y}_t$. If we follow the EI2 strategy (Definition 2), we have

$$(71) = \max_{n \in [N]} \max\left\{\text{ei}\left(\hat{y}_t | m_t^n, \sqrt{c_t^{nn}}\right), \text{ei}\left(-\breve{y}_t | -m_t^n, \sqrt{c_t^{nn}}\right)\right\} \tag{72}$$

and if we follow the UCB2 (Definition 3) strategy, we have

$$(71) = \max\left\{\text{ei}\left(\hat{y}_t \middle| m_t^{n_{ucb}}, \sqrt{c_t^{n_{ucb}n_{ucb}}}\right), \text{ei}\left(-\breve{y}_t \middle| -m_t^{n_{ucb}}, \sqrt{c_t^{n_{ucb}n_{ucb}}}\right)\right\}. \tag{73}$$

Clearly we have (72) $\geq$ (73), hence for both strategies it holds that (71) $\geq$ (73) which concludes the proof.

$\square$

## G.3 Upper Bound for the Left-Hand Side

In analogy with Definition 1, we define

**Definition 8** (Multivariate Expected Improvement). *For a family of jointly Gaussian distributed RVs $(F_n)_{n \in [N]}$ with mean $m \in \mathbb{R}^N$ and covariance $c \in \mathbb{S}_+^N$ and a threshold $\tau \in \mathbb{R}$, we define the multivariate expected improvement as*

$$\text{mei}(\tau|m, c) := \mathbb{E}\left[\max\left\{\max_{n \in [N]} F_n - \tau, 0\right\}\right] \tag{74}$$

$$= \int_{\mathbb{R}^N} \max\left\{\max_{n \in [N]} f_n - \tau, 0\right\} \mathcal{N}(f|m, c)df. \tag{75}$$

**Lemma 6.** *For any instance $(N, T, \mu, \Sigma)$ of the problem defined in Section 2.1, if we follow either the EI2 (Definition 2) or the UCB2 (Definition 3) strategy, we have for any $0 < \beta \leq 1$*

$$\mathbb{E}\left[\beta\hat{\breve{F}} - \hat{\breve{Y}}_t \middle| m_t, c_t, \hat{y}_t, \breve{y}_t\right] \tag{76}$$

$$\leq \beta 2\max\left\{\text{mei}(0|m_t - \hat{y}_t, c_t), \text{mei}(0|\breve{y}_t - m_t, c_t)\right\} + (1-\beta)(-\hat{y}_t + \breve{y}_t)$$

*for any $t \in \{1 : T-1\}, m_t \in \mathbb{R}^N, c_t \in \mathbb{S}_+^N, \hat{y}_t \geq \breve{y}_t \in \mathbb{R}$.*

*Proof.* We have

$$\mathbb{E}\left[\beta\hat{\breve{F}} - \hat{\breve{Y}}_t \middle| m_t, c_t, \hat{y}_t, \breve{y}_t\right] \tag{77}$$

$$= \beta\mathbb{E}\left[\hat{F} - \breve{F} - \hat{y}_t + \breve{y}_t | m_t, c_t\right] + (1-\beta)(-\hat{y}_t + \breve{y}_t) \tag{78}$$

$$= \beta\mathbb{E}\left[\hat{F} - \hat{y}_t | m_t, c_t\right] + \beta\mathbb{E}\left[-\breve{F} + \breve{y}_t | m_t, c_t\right] + (1-\beta)(-\hat{y}_t + \breve{y}_t) \tag{79}$$

$$\leq \beta\mathbb{E}\left[\max\{\hat{F} - \hat{y}_t, 0\} | m_t, c_t\right] + \beta\mathbb{E}\left[\max\{-\breve{F} + \breve{y}_t, 0\} | m_t, c_t\right] + (1-\beta)(-\hat{y}_t + \breve{y}_t) \tag{80}$$

$$= \beta\left(\text{mei}(0|m_t - \hat{y}_t, c_t) + \text{mei}(0|\breve{y}_t - m_t, c_t)\right) + (1-\beta)(-\hat{y}_t + \breve{y}_t) \tag{81}$$

$$\leq \beta 2\max\left\{\text{mei}(0|m_t - \hat{y}_t, c_t), \text{mei}(0|\breve{y}_t - m_t, c_t)\right\} + (1-\beta)(-\hat{y}_t + \breve{y}_t) \tag{82}$$

where by $m_t - \hat{y}_t$ we mean that the scalar $\hat{y}_t$ is subtracted from each element of the vector $m_t$.

$\square$

## G.4 Upper Bound on the Regret

**Theorem 4.** *For any instance $(N, T, \mu, \Sigma)$ of the problem defined in Section 2.1, if we follow either the EI2 (Definition 2) or the UCB2 (Definition 3) strategy, we have for any $\alpha > 0$ and any $0 < \beta \leq 1 - \frac{1}{\sqrt{2\pi}(2\log N)^{3/2}}$*

$$\max_x \left( 2 \frac{\beta \left( \sqrt{2\log N} + \frac{1}{2\log N\sqrt{2\pi}} \right) - x}{\mathrm{ei}\,(x)} \right) \leq \alpha \tag{83}$$

$$\Downarrow$$

$$(1 - e^{-\frac{T-1}{\alpha}})\beta\mathbb{E}\left[ \hat{\check{F}} \right] \leq \mathbb{E}\left[ \hat{\check{Y}} \right] \tag{84}$$

*i.e. the first line implies the second.*

*Proof.* According to Lemma 4, we have

(84)

$$\Uparrow$$

$$\frac{\mathbb{E}\left[ \beta\hat{\check{F}} - \hat{\check{Y}}_t | m_t, c_t, \hat{y}_t, \check{y}_t \right]}{\mathbb{E}\left[ \hat{\check{Y}}_{t+1} - \hat{\check{Y}}_t | m_t, c_t, \hat{y}_t, \check{y}_t \right]} \leq \alpha$$

$$\forall t \in \{1 : T - 1\}, m_t \in \mathbb{R}^N, c_t \in \mathbb{S}_+^N, \hat{y}_t \geq \check{y}_t \in \mathbb{R}. \tag{85}$$

In the following, we will find a simpler expression which implies (85) and therefore (84). Then we will simplify the new expression further, until we finally arrive at (83) through an unbroken chain of implications.

Using the lower bound from Lemma 5 and the upper bound from Lemma 6 we can write

(85)

$$\Uparrow$$

$$\frac{\beta 2\max\left\{\mathrm{mei}(0|m - \hat{y}, c), \mathrm{mei}(0|\check{y} - m, c)\right\} + (1 - \beta)(-\hat{y} + \check{y})}{\max\left\{\mathrm{ei}\left(\hat{y}\Big| m^{n_{ucb}}, \sqrt{c^{n_{ucb}n_{ucb}}}\right), \mathrm{ei}\left(-\check{y}\Big| - m^{n_{ucb}}, \sqrt{c^{n_{ucb}n_{ucb}}}\right)\right\}} \leq \alpha$$

$$\forall m \in \mathbb{R}^N, c \in \mathbb{S}_+^N, \hat{y} \geq \check{y} \in \mathbb{R}, \tag{86}$$

$$n_{ucb} = \underset{n \in [N]}{\mathrm{argmax}} \max\left\{-\hat{y} + m^n + \sqrt{c^{nn}2\log N}, \check{y} - m^n + \sqrt{c^{nn}2\log N}\right\}$$

where we have dropped the time indices, since they are irrelevant here. It is easy to see that all terms in (86) are invariant to a common shift in $m, \hat{y}$ and $\check{y}$. Hence, we can impose a constraint on these variables, without changing the condition. We choose the constraint $\hat{y} = -\check{y}$ to simplify the expression

(86)

$$\Updownarrow$$

$$2\frac{\beta \max\left\{\mathrm{mei}(0|m - \hat{y}, c), \mathrm{mei}(0| - \hat{y} - m, c)\right\} - (1 - \beta)\hat{y}}{\max\left\{\mathrm{ei}\left(\hat{y}\Big| m^{n_{ucb}}, \sqrt{c^{n_{ucb}n_{ucb}}}\right), \mathrm{ei}\left(\hat{y}\Big| - m^{n_{ucb}}, \sqrt{c^{n_{ucb}n_{ucb}}}\right)\right\}} \leq \alpha$$

$$\forall m \in \mathbb{R}^N, c \in \mathbb{S}_+^N, \hat{y} \in \mathbb{R}_{\geq 0}, \tag{87}$$

$$n_{ucb} = \underset{n \in [N]}{\mathrm{argmax}} \max\left\{-\hat{y} + m^n + \sqrt{c^{nn}2\log N}, -\hat{y} - m^n + \sqrt{c^{nn}2\log N}\right\}.$$

Clearly, the denominator is invariant with respect to any sign flips in the elements of $m$. The numerator is maximized if all elements of $m$ have the same sign, no matter if positive or negative. Hence, we can restrict the above conditions to $m$ with positive entries, which means in all maximum operators the left term is active

(87)

$\Updownarrow$

$$2\frac{\beta \operatorname{mei}(0|m-\hat{y},c) - (1-\beta)\hat{y}}{\operatorname{ei}\left(\hat{y}\middle| m^{n_{ucb}}, \sqrt{c^{n_{ucb}n_{ucb}}}\right)} \leq \alpha$$

$$\forall m \in \mathbb{R}_{\geq 0}^N, c \in \mathbb{S}_+^N, \hat{y} \in \mathbb{R}_{\geq 0}, n_{ucb} = \operatorname*{argmax}_{n \in [N]}\left(m^n + \sqrt{c^{nn}2\log N}\right). \qquad (88)$$

Inserting the bound from Lemma 12 we have

(88)

$\Uparrow$

$$2\frac{\beta\left(\max\left\{\max_{n\in[N]}(m^n - \hat{y} + \sqrt{c^{nn}2\log N}),0\right\} + \frac{\max_{n\in[N]}\sqrt{c^{nn}}}{2\sqrt{2\pi}\log(N)}\right) - (1-\beta)\hat{y}}{\operatorname{ei}\left(\hat{y}\middle| m^{n_{ucb}}, \sqrt{c^{n_{ucb}n_{ucb}}}\right)} \leq \alpha$$

$$\forall m \in \mathbb{R}_{\geq 0}^N, c \in \mathbb{S}_+^N, \leq \alpha, \hat{y} \in \mathbb{R}_{\geq 0}, n_{ucb} = \operatorname*{argmax}_{n \in [N]}\left(m^n + \sqrt{c^{nn}2\log N}\right) \qquad (89)$$

$\Updownarrow$

$$2\frac{\beta\left(\max\left\{m^{n_{ucb}} - \hat{y} + \sqrt{c^{n_{ucb}n_{ucb}}2\log N},0\right\} + \frac{\max_{n\in[N]}\sqrt{c^{nn}}}{2\sqrt{2\pi}\log(N)}\right) - (1-\beta)\hat{y}}{\operatorname{ei}\left(\hat{y}\middle| m^{n_{ucb}}, \sqrt{c^{n_{ucb}n_{ucb}}}\right)} \leq \alpha$$

$$\forall m \in \mathbb{R}_{\geq 0}^N, c \in \mathbb{S}_+^N, \leq \alpha, \hat{y} \in \mathbb{R}_{\geq 0}, n_{ucb} = \operatorname*{argmax}_{n \in [N]}\left(m^n + \sqrt{c^{nn}2\log N}\right). \qquad (90)$$

It holds for any $n \in [N]$ that $m^n \geq 0$ and

$$m^{n_{ucb}} + \sqrt{c^{n_{ucb}n_{ucb}}}\sqrt{2\log N} \geq m^n + \sqrt{c^{nn}2\log N},$$

from which it follows that

$$\frac{m^{n_{ucb}}}{\sqrt{2\log N}} + \sqrt{c^{n_{ucb}n_{ucb}}} \geq \sqrt{c^{nn}}. \qquad (91)$$

Using this fact we can write

(90)

$\Uparrow$

$$2\frac{\beta\left(\max\{m^{n_{ucb}} - \hat{y} + \sqrt{c^{n_{ucb}n_{ucb}}}\sqrt{2\log N},0\} + \frac{\sqrt{c^{n_{ucb}n_{ucb}}}}{2\log N\sqrt{2\pi}} + \frac{m^{n_{ucb}}}{\sqrt{2\pi}(2\log N)^{3/2}}\right) - (1-\beta)\hat{y}}{\operatorname{ei}\left(\hat{y}\middle| m^{n_{ucb}}, \sqrt{c^{n_{ucb}n_{ucb}}}\right)} \leq \alpha$$

$$\forall m \in \mathbb{R}_{\geq 0}^N, c \in \mathbb{S}_+^N, \leq \alpha, \hat{y} \in \mathbb{R}_{\geq 0}, n_{ucb} = \operatorname*{argmax}_{n \in [N]}\left(m^n + \sqrt{c^{nn}2\log N}\right). \qquad (92)$$

For any $\beta \leq 1 - \frac{1}{1+\sqrt{2\pi}(2\log N)^{3/2}} \leq 1 - \frac{1}{\sqrt{2\pi}(2\log N)^{3/2}}$ we have

(92)

$\Uparrow$

$$2\frac{\beta\left(\max\{m^{n_{ucb}} - \hat{y} + \sqrt{c^{n_{ucb}n_{ucb}}}\sqrt{2\log N}, 0\} + \frac{\sqrt{c^{n_{ucb}n_{ucb}}}}{2\log N\sqrt{2\pi}}\right) - (1-\beta)(\hat{y} - m^{n_{ucb}})}{\mathrm{ei}\left(\hat{y}\Big| m^{n_{ucb}}, \sqrt{c^{n_{ucb}n_{ucb}}}\right)} \leq \alpha$$

$$\forall m \in \mathbb{R}_{\geq 0}^N, c \in \mathbb{S}_+^N, \leq \alpha, \hat{y} \in \mathbb{R}_{\geq 0}, n_{ucb} = \operatorname*{argmax}_{n\in[N]}\left(m^n + \sqrt{c^{nn}2\log N}\right). \tag{93}$$

According to Definition 1, we have

$$\mathrm{ei}\left(\hat{y}\Big|\mu, \sqrt{c^{n_{ucb}n_{ucb}}}\right) = \sqrt{c^{n_{ucb}n_{ucb}}}\,\mathrm{ei}\left(\frac{\hat{y} - m^{n_{ucb}}}{\sqrt{c^{n_{ucb}n_{ucb}}}}\right).$$

Using this fact, we obtain

(93)

$\Updownarrow$

$$2\frac{\beta\left(\max\left\{\frac{m^{n_{ucb}} - \hat{y}}{\sqrt{c^{n_{ucb}n_{ucb}}}} + \sqrt{2\log N}, 0\right\} + \frac{1}{2\log N\sqrt{2\pi}}\right) - (1-\beta)\frac{\hat{y} - m^{n_{ucb}}}{\sqrt{c^{n_{ucb}n_{ucb}}}}}{\mathrm{ei}\left(\frac{\hat{y} - m^{n_{ucb}}}{\sqrt{c^{n_{ucb}n_{ucb}}}}\right)} \leq \alpha$$

$$\forall m \in \mathbb{R}_{\geq 0}^N, c \in \mathbb{S}_+^N, \leq \alpha, \hat{y} \in \mathbb{R}_{\geq 0}, n_{ucb} = \operatorname*{argmax}_{n\in[N]}\left(m^n + \sqrt{c^{nn}2\log N}\right). \tag{94}$$

Defining $x := \frac{\hat{y} - m^{n_{ucb}}}{\sqrt{c^{n_{ucb}n_{ucb}}}}$ we can simplify this condition as

(94)

$\Updownarrow$

$$2\frac{\beta\left(\max\left\{\sqrt{2\log N} - x, 0\right\} + \frac{1}{2\log N\sqrt{2\pi}}\right) - (1-\beta)x}{\mathrm{ei}(x)} \leq \alpha \qquad \forall x \in \mathbb{R}. \tag{95}$$

For any $x \geq \sqrt{2\log N}$ the numerator of the left hand side is negative

$$\beta\left(\frac{1}{2\log N\sqrt{2\pi}}\right) - (1-\beta)x \leq \beta\left(\frac{1}{2\log N\sqrt{2\pi}}\right) - (1-\beta)\sqrt{2\log N} \tag{96}$$

$$\leq (1 - \frac{1}{\sqrt{2\pi}(2\log N)^{3/2}})\left(\frac{1}{2\log N\sqrt{2\pi}}\right) - \frac{1}{\sqrt{2\pi}(2\log N)} \tag{97}$$

$$= -\frac{1}{2\log N\sqrt{2\pi}}\left(\frac{1}{2\log N\sqrt{2\pi}}\right) \tag{98}$$

$$\leq 0 \tag{99}$$

and since $\alpha$ and the denominator are both positive, (95) is satisfied. Hence, we only need to consider the case where $x \leq \sqrt{2\log N}$ and can therefore write

(95)

$\Updownarrow$

$$2\frac{\beta\left(\sqrt{2\log N} + \frac{1}{2\log N\sqrt{2\pi}}\right) - x}{\mathrm{ei}(x)} \leq \alpha \quad \forall x \leq \sqrt{2\log N}. \tag{100}$$

From this chain of implications and Lemma 4 the result of Theorem 4 follows. $\qquad\square$

Finally, using the previous results, we can obtain the desired bound on the regret (Theorem 1), which we restate here for convenience:

**Theorem 1.** *For any instance $(N, T, \mu, \Sigma)$ of the problem defined in Section 2.1 with $N \geq T \geq 500$, if we follow either the EI2 (Definition 2) or the UCB2 (Definition 3) strategy, we have*

$$\frac{\mathbb{E}\left[\hat{\hat{F}}\right] - \mathbb{E}\left[\hat{\hat{Y}}\right]}{\mathbb{E}\left[\hat{\hat{F}}\right]} \leq 1 - \left(1 - T^{-\frac{1}{2\sqrt{\pi}}}\right)\sqrt{\frac{\log(T) - \log\left(3\log^{\frac{3}{2}}(T)\right)}{\log(N)}}. \tag{14}$$

*This guarantees that the expected difference between the maximum and minimum retrieved function value achieves a certain ratio with respect to the expected difference between the global maximum and the global minimum.*

*Proof.* We can rewrite Theorem 4 as

$$\left(1 - e^{-\frac{T}{\max_x\left(2\frac{\beta\left(\sqrt{2\log N} + \frac{1}{2\log N\sqrt{2\pi}}\right) - x}{\text{ei}(x)}\right)}}\right)\beta \leq \frac{\mathbb{E}\left[\hat{\hat{Y}}\right]}{\mathbb{E}\left[\hat{\hat{F}}\right]} \tag{101}$$

$$\forall N \geq T \geq 500, \mu, \Sigma, 0 < \beta \leq 1 - \frac{1}{\sqrt{2\pi}\left(2\log N\right)^{3/2}}$$

where we have restricted the inequality to $N \geq T \geq 500$, a condition we need later . What is left to be done is to simplify this bound, such that it becomes interpretable. We can rewrite the above as

$$\left(1 - e^{-\frac{T}{\max_x\left(2\frac{\beta\left(\frac{1}{\sqrt{2\pi}a^2} + a\right) - x}{\text{ei}(x)}\right)}}\right)\beta \leq \frac{\mathbb{E}\left[\hat{\hat{Y}}\right]}{\mathbb{E}\left[\hat{\hat{F}}\right]} \tag{102}$$

$$\forall N \geq T \geq 500, \mu, \Sigma, 0 < \beta \leq 1 - \frac{1}{\sqrt{2\pi}a^3}, a = \sqrt{2\log N}.$$

**Choosing a $\beta$**

We choose

$$\beta = \frac{b - \frac{\text{ei}(b)}{\text{ei}'(b)}}{\frac{1}{\sqrt{2\pi}a^2} + a} \tag{103}$$

where $\text{ei}'$ is the derivative of ei, and

$$b = \sqrt{\sqrt{2\log T}^2 - 2\log\left(3\, 2^{-3/2}\sqrt{2\log T}^3\right)}. \tag{104}$$

Before we can continue, we need to show that this $\beta$ satisfies the condition in (102). First of all, from (104) and the conditions in (102) we can derive some relations which will be useful later on

$$2.17 \leq b \leq \sqrt{a^2 - 2\log\left(3\, 2^{-3/2}a^3\right)} \leq a. \tag{105}$$

It is easy to see that $0 < \beta$ holds, it remains to be shown that

$$\beta \leq 1 - \frac{1}{\sqrt{2\pi}a^3}. \tag{106}$$

Inserting the definition of $\beta$ (103) and simplifying we have

$$\frac{b - \frac{\text{ei}(b)}{\text{ei}'(b)}}{\frac{1}{\sqrt{2\pi}a^2} + a} \leq 1 - \frac{1}{\sqrt{2\pi}a^3} \tag{107}$$

$$b - \frac{\text{ei}(b)}{\text{ei}'(b)} \leq a - \frac{1}{2\pi a^5}. \tag{108}$$

Using Definition 1 and the lower bound in Lemma 10, we obtain a sufficient condition

$$\frac{b^3}{b^2 - 1} \le a - \frac{1}{2\pi a^5}. \tag{109}$$

Since we have $b \le a$ (see (105)) we obtain the sufficient condition

$$\frac{1}{2\pi b^5} + \frac{b^3}{b^2 - 1} \le a. \tag{110}$$

From $2.17 \le b$ (see (105)) it follows that $2\pi b^5 \ge b^2 - 1$, and hence we can further simplify to obtain a sufficient condition

$$b + \frac{1}{b - 1} \le a. \tag{111}$$

Given (105) it is easy to see that both sides are positive, hence we can square each side to obtain

$$b^2 + \frac{2b}{b - 1} + \frac{1}{(b - 1)^2} \le a^2. \tag{112}$$

Now we will show that this condition is satisfied by (104). We have from (105)

$$b \le \sqrt{a^2 - 2\log\left(3\,2^{-3/2}a^3\right)} \tag{113}$$

$$2\log\left(3\,2^{-3/2}a^3\right) + b^2 \le a^2 \tag{114}$$

and since $b \le a$, we have

$$2\log\left(3\,2^{-3/2}b^3\right) + b^2 \le a^2. \tag{115}$$

This implies (112), to see this we use the above inequality to bound $a^2$ in (112)

$$b^2 + \frac{2b}{b - 1} + \frac{1}{(b - 1)^2} \le 2\log\left(3\,2^{-3/2}b^3\right) + b^2 \tag{116}$$

$$\frac{2b}{b - 1} + \frac{1}{(b - 1)^2} \le 2\log\left(3\,2^{-3/2}b^3\right). \tag{117}$$

It is easy to verify that for any $b \ge 2.17$, the left-hand side is decreasing and the right-hand side is increasing. Hence, it is sufficient to show that it holds for $b = 2.17$ which is easily done by evaluating at that value. This concludes the proof that our choice of $\beta$ (103) satisfies the conditions in (102). We can now insert this value to obtain

$$\left(1 - e^{-\frac{T}{\max_x\left(2\frac{b - \frac{\mathrm{ei}(b)}{\mathrm{ei}'(b)} - x}{\mathrm{ei}(x)}\right)}}\right) \frac{b - \frac{\mathrm{ei}(b)}{\mathrm{ei}'(b)}}{\frac{1}{\sqrt{2\pi a^2}} + a} \le \frac{\mathbb{E}\left[\hat{Y}\right]}{\mathbb{E}\left[\hat{F}\right]} \tag{118}$$

$$\forall N \ge T \ge 500, \bar{\mu}, \Sigma, a = \sqrt{2\log N}, b = \sqrt{\sqrt{2\log T}^2 - 2\log\left(3\,2^{-3/2}\sqrt{2\log T}^3\right)}$$

**Optimizing for $x$**

Here we show that

$$\max_x\left(2\frac{b - \frac{\mathrm{ei}(b)}{\mathrm{ei}'(b)} - x}{\mathrm{ei}(x)}\right) \le -\frac{2}{\mathrm{ei}'(b)}. \tag{119}$$

We have

$$\max_x\left(2\frac{b - \frac{\mathrm{ei}(b)}{\mathrm{ei}'(b)} - x}{\mathrm{ei}(x)}\right) = \max_\delta\left(2\frac{b - \frac{\mathrm{ei}(b)}{\mathrm{ei}'(b)} - b - \delta}{\mathrm{ei}(b + \delta)}\right) \tag{120}$$

$$= \max_\delta\left(2\frac{\mathrm{ei}(b) + \delta\mathrm{ei}'(b)}{-\mathrm{ei}'(b)\,\mathrm{ei}(b + \delta)}\right). \tag{121}$$

Since the ei function is convex, we have $\mathrm{ei}(b) + \delta \mathrm{ei}'(b) \leq \mathrm{ei}(b + \delta)$. Using this and the fact that the denominator is positive (since ei is always positive and $\mathrm{ei}'$ is always negative), we have

$$\max_x \left( 2 \frac{b - \frac{\mathrm{ei}(b)}{\mathrm{ei}'(b)} - x}{\mathrm{ei}(x)} \right) \leq -\frac{2}{\mathrm{ei}'(b)}. \tag{122}$$

Inserting this result into (118), we obtain

$$\left( 1 - e^{\frac{T\mathrm{ei}'(b)}{2}} \right) \frac{b - \frac{\mathrm{ei}(b)}{\mathrm{ei}'(b)}}{\frac{1}{\sqrt{2\pi}a^2} + a} \leq \frac{\mathbb{E}\left[ \hat{\hat{Y}} \right]}{\mathbb{E}\left[ \hat{\hat{F}} \right]} \tag{123}$$

$$\forall N \geq T \geq 500, \mu, \Sigma, a = \sqrt{2 \log N}, b = \sqrt{\sqrt{2 \log T}^2 - 2 \log \left( 3 \, 2^{-3/2} \sqrt{2 \log T}^3 \right)}.$$

**Simplifying the bound further**

Now, all that is left to do is to simplify this bound a bit further.

**Bounding the left factor**

First of all, we show that the left factor satisfies

$$1 - e^{\frac{1}{2}T\mathrm{ei}'(b)} \geq 1 - T^{-\frac{1}{2\sqrt{\pi}}}. \tag{124}$$

Using Definition 1 and the lower bound from Lemma 10, we obtain

$$1 - e^{\frac{1}{2}T\mathrm{ei}'(b)} \geq 1 - e^{-\frac{\left( b^2 - 1 \right) e^{-\frac{b^2}{2}} T}{2\sqrt{2\pi}b^3}} \tag{125}$$

$$\geq 1 - e^{-\frac{e^{-\frac{b^2}{2}} T}{3\sqrt{2\pi}b}} \tag{126}$$

where the second inequality is easily seen to hold true, since we have $b \geq 2.17$. Inserting (104), and bounding further we obtain

$$1 - e^{-\frac{e^{-\frac{b^2}{2}} T}{3\sqrt{2\pi}b}} = 1 - e^{-\frac{\log^{\frac{3}{2}}(T)}{2\sqrt{\pi}\sqrt{\log(T) - \log\left( 3 \log^{\frac{3}{2}}(T) \right)}}} \tag{127}$$

$$\geq 1 - e^{-\frac{\log^{\frac{3}{2}}(T)}{2\sqrt{\pi}\sqrt{\log(T)}}} \tag{128}$$

$$= 1 - T^{-\frac{1}{2\sqrt{\pi}}} \tag{129}$$

from which (124) follows.

**Bounding the right factor**

Now we will show that the right factor from (123) satisfies

$$\frac{b - \frac{\mathrm{ei}(b)}{\mathrm{ei}'(b)}}{\frac{1}{\sqrt{2\pi}a^2} + a} \geq \sqrt{\frac{\log(T) - \log\left( 3 \log^{\frac{3}{2}}(T) \right)}{\log(N)}}. \tag{130}$$

Using Definition 1 and the upper bound from Lemma 10, we obtain

$$\frac{b - \frac{\mathrm{ei}(b)}{\mathrm{ei}'(b)}}{\frac{1}{\sqrt{2\pi}a^2} + a} \geq \frac{2\sqrt{\pi}a^2 b^5}{\left( 2\sqrt{\pi}a^3 + \sqrt{2} \right) \left( b^4 - b^2 + 3 \right)}. \tag{131}$$

Now, to lower bound this further, we show that

$$\left( 2\sqrt{\pi}a^3 + \sqrt{2} \right) \left( b^4 - b^2 + 3 \right) \leq \left( 2\sqrt{\pi}a^3 \right) b^4. \tag{132}$$

Equivalently, we can show that

$$-2\sqrt{\pi}a^3b^2 + 6\sqrt{\pi}a^3 + \sqrt{2}b^4 - \sqrt{2}b^2 + 3\sqrt{2} \le 0. \tag{133}$$

Since $b \ge 2.17$, we have $3\sqrt{2} - \sqrt{2}b^2 \le 0$, and hence

$$-2\sqrt{\pi}a^3b^2 + 6\sqrt{\pi}a^3 + \sqrt{2}b^4 \le 0 \tag{134}$$

is a sufficient condition. The derivative of the left-hand side

$$4\sqrt{2}b^2 - 4\sqrt{\pi}a^3 \le 4\sqrt{2}a^2 - 4\sqrt{\pi}a^3 \tag{135}$$

$$= (4\sqrt{2} - 4\sqrt{\pi}a)a^2 \tag{136}$$

is always negative (since $a \ge 2.17$). Hence, it is sufficient to show that (134) holds for the minimal $b = 2.17$, which can easily be verified to hold true for any $a \ge 2.17$. Hence, we have shown that (132) holds, and inserting it into (131), we obtain

$$\frac{b - \frac{\text{ei}(b)}{\text{ei}'(b)}}{\frac{1}{\sqrt{2\pi}a^2} + a} \ge \frac{2\sqrt{\pi}a^2b^5}{(2\sqrt{\pi}a^3)\,b^4} \tag{137}$$

$$= \frac{b}{a} \tag{138}$$

$$= \sqrt{\frac{\log(T) - \log\left(3\log^{\frac{3}{2}}(T)\right)}{\log(N)}}. \tag{139}$$

**Final bound**

Finally, inserting (124) and (130) into (123), we obtain

$$\left(1 - T^{-\frac{1}{2\sqrt{\pi}}}\right)\sqrt{\frac{\log(T) - \log\left(3\log^{\frac{3}{2}}(T)\right)}{\log(N)}} \le \frac{\mathbb{E}\left[\overset{\circ}{\check{Y}}\right]}{\mathbb{E}\left[\overset{\circ}{\check{F}}\right]} \quad \forall N \ge T \ge 500, \mu, \Sigma. \tag{140}$$

This implies straightforwardly the bound on the regret

$$\frac{\mathbb{E}\left[\overset{\circ}{\check{F}}\right] - \mathbb{E}\left[\overset{\circ}{\check{Y}}\right]}{\mathbb{E}\left[\overset{\circ}{\check{F}}\right]} \le 1 - \left(1 - T^{-\frac{1}{2\sqrt{\pi}}}\right)\sqrt{\frac{\log(T) - \log\left(3\log^{\frac{3}{2}}(T)\right)}{\log(N)}} \quad \forall N \ge T \ge 500, \mu, \Sigma. \tag{141}$$

$\square$

# H   Proof of Theorem 3

The proof of Theorem 3 is very similar to the one of Theorem 1 in Appendix G. Here we discuss the parts that are different.

**Lemma 7.** *For any problem of the type defined in Section 2.1, we have for any $\alpha, \beta > 0$*

$$\beta\mathbb{E}\left[\overset{\circ}{\check{F}}\right] - \mathbb{E}\left[\hat{Y}_t - \check{F}\right] \le \alpha\left(\mathbb{E}\left[\hat{Y}_{t+1} - \hat{Y}_t\right]\right) \quad \forall t \in \{1 : T - 1\} \tag{142}$$

$$\Downarrow$$

$$(1 - e^{-\frac{T-1}{\alpha}})\beta\mathbb{E}\left[\overset{\circ}{\check{F}}\right] \le \mathbb{E}\left[\hat{Y}_T - \check{F}\right]. \tag{143}$$

*i.e. the first inequality implies the second inequality.*

*Proof.* This result is obtained from Lemma 3 by replacing $\overset{\circ}{\hat{Y}}_t$ with $\hat{Y}_t - \check{F}$. It is easy to see that the proof of Lemma 3 goes through with this change.

$\square$

Hence, if for some $\alpha, \beta > 0$ we can show that (142) holds, Lemma 7 yields a lower bound on the expected utility.

**Lemma 8.** *For any problem of the type defined in Section 2.1 we have for any $\alpha, \beta > 0$*

$$\mathbb{E}\left[\beta\hat{\check{F}} - (\hat{Y}_t - \check{F})|m_t, c_t, \hat{y}_t\right] \leq \alpha\mathbb{E}\left[\hat{Y}_{t+1} - \hat{Y}_t|m_t, c_t, \hat{y}_t\right] \tag{144}$$

$$\forall t \in \{1 : T - 1\}, m_t \in \mathbb{R}^N, c_t \in \mathbb{S}_+^N, \hat{y}_t \in \mathbb{R}$$

$$\Downarrow$$

$$(1 - e^{-\frac{T-1}{\alpha}})\beta\mathbb{E}\left[\hat{\check{F}}\right] \leq \mathbb{E}\left[\hat{Y} - \check{F}\right]. \tag{145}$$

*Proof.* The proof follows the same logic as the one from Lemma 4.

$\square$

**Theorem 5.** *For any instance $(N, T, \mu, \Sigma)$ of the problem defined in Section 2.1, if we follow either the EI (Definition 6) or the UCB (Definition 7) strategy, we have for any $\alpha > 0$ and any $0 < \beta \leq 1 - \frac{1}{\sqrt{2\pi}(2\log N)^{3/2}}$*

$$\max_x \left(2\frac{\beta\left(\sqrt{2\log N} + \frac{1}{2\log N\sqrt{2\pi}}\right) - x}{\mathrm{ei}\,(x)}\right) \leq \alpha \tag{146}$$

$$\Downarrow$$

$$(1 - e^{-\frac{T-1}{\alpha}})\beta\mathbb{E}\left[\hat{\check{F}}\right] \leq \mathbb{E}\left[\hat{Y} - \check{F}\right] \tag{147}$$

*i.e. the first line implies the second.*

*Proof.* According to Lemma 8, we have

$$(147)$$

$$\Uparrow$$

$$\frac{\mathbb{E}\left[\beta\hat{\check{F}} - (\hat{Y}_t - \check{F})|m_t, c_t, \hat{y}_t\right]}{\mathbb{E}\left[\hat{Y}_{t+1} - \hat{Y}_t|m_t, c_t, \hat{y}_t\right]} \leq \alpha$$

$$\forall t \in \{1 : T - 1\}, m_t \in \mathbb{R}^N, c_t \in \mathbb{S}_+^N, \hat{y}_t \in \mathbb{R}. \tag{148}$$

Rearranging terms and plugging in the known variable $\hat{y}_t$ we have

$$(148)$$

$$\Updownarrow \tag{149}$$

$$\frac{\beta\mathbb{E}\left[\hat{F} - \hat{y}_t|m_t, c_t\right] - (1 - \beta)\mathbb{E}\left[\hat{y}_t - \check{F}|m_t, c_t\right]}{\mathbb{E}\left[\hat{Y}_{t+1} - \hat{y}_t|m_t, c_t, \hat{y}_t\right]} \leq \alpha$$

$$\forall t \in \{1 : T - 1\}, m_t \in \mathbb{R}^N, c_t \in \mathbb{S}_+^N, \hat{y}_t \in \mathbb{R}. \tag{150}$$

Since the expected minimum function value $\check{F}$ is no larger than the smallest mean $\check{m}_t$, we have

$$(150)$$

$$\Uparrow \tag{151}$$

$$\frac{\beta\mathbb{E}\left[\hat{F} - \hat{y}_t|m_t, c_t\right] - (1 - \beta)(\hat{y}_t - \check{m}_t)}{\mathbb{E}\left[\hat{Y}_{t+1} - \hat{y}_t|m_t, c_t, \hat{y}_t\right]} \leq \alpha$$

$$\forall t \in \{1 : T - 1\}, m_t \in \mathbb{R}^N, c_t \in \mathbb{S}_+^N, \hat{y}_t \in \mathbb{R}. \tag{152}$$

Since these terms are invariant to a common shift in all variables, we can assume $\check{m}_t = 0$ without loss of generality, and hence

$$(152)$$

$$\Updownarrow \tag{153}$$

$$\frac{\beta \mathbb{E}\left[\hat{F} - \hat{y}_t | m_t, c_t\right] - (1 - \beta)\hat{y}_t}{\mathbb{E}\left[\hat{Y}_{t+1} - \hat{y}_t | m_t, c_t, \hat{y}_t\right]} \le \alpha$$

$$\forall t \in \{1 : T - 1\}, m_t \in \mathbb{R}_{\ge 0}^N, c_t \in \mathbb{S}_+^N, \hat{y}_t \in \mathbb{R}_{\ge 0}. \tag{154}$$

We have

$$\mathbb{E}\left[\hat{F} - \hat{y}_t | m_t, c_t\right] \le \mathbb{E}\left[\max\{\hat{F} - \hat{y}_t, 0\} | m_t, c_t\right] \tag{155}$$

$$= \mathbb{E}\left[\max\{\hat{F}, 0\} | m_t - \hat{y}_t, c_t\right] \tag{156}$$

$$= \mathrm{mei}(0 | m_t - \hat{y}_t, c_t) \tag{157}$$

with mei as defined in Definition 8. In addition, we have

$$\mathbb{E}\left[\hat{Y}_{t+1} - \hat{y}_t | m_t, c_t, \hat{y}_t\right] = \mathbb{E}\left[\max\left\{\hat{Y}_{t+1} - \hat{y}_t, 0\right\} | m_t, c_t, \hat{y}_t\right] \tag{158}$$

$$\ge \mathrm{ei}\left(\hat{y}_t | m_t^{n_{ucb}}, \sqrt{c_t^{n_{ucb}n_{ucb}}}\right) \tag{159}$$

with ei as defined in Definition 1. The equality follows from the fact that we know that $\hat{Y}_{t+1} \ge \hat{y}_t$. The inequality follows due to a similar argument as the one in the proof of Lemma 5.

Substituting these terms and dropping the time index, we have

$$(154)$$

$$\Uparrow$$

$$\frac{\beta \, \mathrm{mei}(0 | m - \hat{y}, c) - (1 - \beta)\hat{y}}{\mathrm{ei}\left(\hat{y} \middle| m^{n_{ucb}}, \sqrt{c^{n_{ucb}n_{ucb}}}\right)} \le \alpha$$

$$\forall m \in \mathbb{R}_{\ge 0}^N, c \in \mathbb{S}_+^N, \hat{y} \in \mathbb{R}_{\ge 0} \tag{160}$$

$$n_{ucb} = \underset{n \in [N]}{\arg\max} \max\left(m^n + \sqrt{c^{nn}2\log N}\right). \tag{161}$$

Note that this condition is implied by (88), hence the rest of the proof is identical to the one from Theorem 4. $\qquad\square$

Finally, using the previous results, we can obtain the desired bound on the regret (Theorem 3), which we restate here for convenience:

**Theorem 3.** *For any instance $(N, T, \mu, \Sigma)$ of the problem defined in Section 2.1 with $N \ge T \ge 500$, if we follow either the **EI** (Definition 6) or the **UCB strategy** (Definition 7) we have*

$$\frac{\mathbb{E}\left[\hat{\check{F}}\right] - \mathbb{E}\left[\hat{\boldsymbol{Y}} - \check{\boldsymbol{F}}\right]}{\mathbb{E}\left[\hat{\check{F}}\right]} \le 1 - \left(1 - T^{-\frac{1}{2\sqrt{\pi}}}\right)\sqrt{\frac{\log(T) - \log\left(3\log^{\frac{3}{2}}(T)\right)}{\log(N)}}. \tag{33}$$

*Proof.* The proof is identical to the one from Theorem 1. We simply substitute $\hat{\check{Y}}$ with $\hat{Y} - \check{F}$ and the proof goes through unchanged otherwise. $\qquad\square$

# I  Proof of Lemma 1

For convenience, we restate Lemma 1 before the proof:

**Lemma 1** (Lower Bound). *For the instance of the problem defined in Section 2.1 with $\mu = \mathbf{0}$ and $\Sigma = \mathbf{I}$, the following lower bound on the regret holds for the optimal strategy: $\forall \epsilon > 0 \; \exists K : \forall N \geq T \geq K$ :*

$$\text{normreg} \geq 1 - \frac{\sqrt{\log T}}{\sqrt{\log N}} - \epsilon. \tag{22}$$

*Proof.* Since the bandits are i.i.d., an optimal strategy is to select arms uniformly at random (without replacement), which means that at each step we observe an i.i.d. sample. It is known that for i.i.d. standard normal random variables $X_1, ..., X_K$, we have asymptotically

$$\mathbb{E}[\max\{X_1, ..., X_K\}] \sim \sqrt{2 \log K} \quad (K \to \infty), \tag{162}$$

see e.g. Massart [2007] page 66. It follows that $\forall \epsilon > 0 \; \exists K : \forall N \geq T \geq K$ :

$$\frac{\mathbb{E}[\hat{Y}]}{\mathbb{E}[\hat{F}]} \leq \frac{\sqrt{2 \log T}}{\sqrt{2 \log N}} + \epsilon \tag{163}$$

from which the desired result follows straightforwardly. $\qquad\square$

# J  Proof of Lemma 2

For convenience, we restate Lemma 2 before the proof:

**Lemma 2.** *For any optimization policy which does not depend on the prior $(\mu, \Sigma)$, there exists an instance of the problem defined in Section 2.1 where*

$$\text{normreg} \geq 1 - \frac{T}{N}, \tag{24}$$

*which is clearly worse for $T \ll N$ than the bound in Corollary 1.*

*Proof.* Suppose we have an optimization policy, and the prior mean $\mu$ and covariance $\Sigma$ are both zero, i.e. the GP is zero everywhere. This will induce a distribution over the actions $A_{1:T}$ taken by the policy. Since there are $N$ possible actions and the policy is allowed to pick $T$ of them, there is at least one action $K$ which has a probability of no more than $T/N$ of being chosen. Now suppose that we set the prior covariance $\Sigma_{KK}$ to a nonzero value, while maintaining everything else zero. Note that this will not change the distribution over the actions taken by the policy unless it happens to pick $K$, hence the probability of action $K$ being selected does not change. Since the observed maximum $\hat{Y}$ is $\hat{F}$ if action $K$ is selected by the policy and zero otherwise, we have

$$\mathbb{E}[\hat{Y}] \leq \mathbb{E}[\hat{F}] \frac{T}{N} \tag{164}$$

from which (24) follows straightforwardly. $\qquad\square$

# K  Proof of Theorem 2

For convenience, we restate Theorem 2 before the proof:

**Theorem 2.** *For any centered Gaussian Process $(G_a)_{a \in \mathcal{A}}$, where $\mathcal{A}$ is the $D$-dimensional unit cube, with kernel $k$ such that*

$$|k(x, x) - k(x, y)| \leq L_k \|x - y\|_\infty \quad \forall x, y \in \mathcal{A} \tag{30}$$

$$\sqrt{k(x, x)} \leq \sigma \quad \forall x \in \mathcal{A} \tag{31}$$

*we obtain the following bound on the regret, if we follow the EI2 (Definition 2) or the UCB2 (Definition 3) strategy:*

$$\mathbb{E}\left[\sup_{a\in\mathcal{A}} G(a) - \hat{Y}\right] \leq \sqrt{\frac{2\log(L_k)}{T^{1/D}}} \left(2\sqrt{\log\left(2\left\lceil\frac{L_k}{\log(L_k)}T^{1/D}\right\rceil^D\right)} + 15\sqrt{D}\right) +$$

$$\sqrt{2}\sigma\left(\sqrt{D\log\left(\left\lceil\frac{L_k}{\log(L_k)}T^{1/D}\right\rceil\right)} - \left(1 - T^{-\frac{1}{2\sqrt{\pi}}}\right)\sqrt{\log\left(\frac{T}{3\log^{\frac{3}{2}}(T)}\right)}\right). \quad (32)$$

*This bound also holds when restricting EI2 or UCB2 to a uniform grid on the domain $\mathcal{A}$, where each side is divided into $\left\lceil\frac{L_k}{\log(L_k)}T^{1/D}\right\rceil$ segments. Finally, this bound converges to 0 as $T \to \infty$.*

*Proof.* The idea here is to pre-select a set of $N$ points at locations $X_{1:N}$ on a grid and then sub-select points from this set during runtime using EI2 (Definition 2) or UCB2 (Definition 3). The regret of this strategy can be bounded by

$$\mathbb{E}\left[\sup_{a\in\mathcal{A}} G(a) - \hat{Y}\right] \leq \mathbb{E}\left[\sup_{a\in\mathcal{A}} G(a) - \max_{i\in[N]} G_{X_i}\right] + \mathbb{E}\left[\max_{i\in[N]} G_{X_i} - \hat{Y}\right]. \quad (165)$$

A bound on the first term is given by the main result in Grünewälder et al. [2010]:

$$\mathbb{E}\left[\sup_{a\in\mathcal{A}} G(a) - \max_{i\in[N]} G_{X_i}\right] \leq \sqrt{\frac{2L_k}{\lfloor N^{1/D}\rfloor}}\left(2\sqrt{\log(2N)} + 15\sqrt{D}\right). \quad (166)$$

A bound on the second term can be derived straightforwardly from Corollary 1:

$$\mathbb{E}\left[\max_{i\in[N]} G_{A_i} - \hat{Y}\right] \leq \left(1 - \left(1 - T^{-\frac{1}{2\sqrt{\pi}}}\right)\sqrt{\frac{\log\left(\frac{T}{3\log^{\frac{3}{2}}(T)}\right)}{\log(N)}}\right)\max_{i\in[N]} G_{A_i} \quad (167)$$

$$\leq \sqrt{2}\sigma\left(\sqrt{\log(N)} - \left(1 - T^{-\frac{1}{2\sqrt{\pi}}}\right)\sqrt{\log\left(\frac{T}{3\log^{\frac{3}{2}}(T)}\right)}\right). \quad (168)$$

where we have used the inequality $\max_{i\in[N]} G_{A_i} \leq \sigma\sqrt{2\log(N)}$ (see Massart [2007], Lemma 2.3).

Choosing $N = \left\lceil\frac{L_k}{\log(L_k)}T^{1/D}\right\rceil^D$ we obtain

$$\mathbb{E}\left[\sup_{a\in\mathcal{A}} G(a) - \max_{i\in[N]} G_{X_i}\right] \leq \sqrt{\frac{2L_k}{\left\lceil\frac{L_k}{\log(L_k)}T^{1/D}\right\rceil}}\left(2\sqrt{\log\left(2\left\lceil\frac{L_k}{\log(L_k)}T^{1/D}\right\rceil^D\right)} + 15\sqrt{D}\right)$$

$$(169)$$

$$\leq\sqrt{\frac{2\log(L_k)}{T^{1/D}}}\left(2\sqrt{\log\left(2\left\lceil\frac{L_k}{\log(L_k)}T^{1/D}\right\rceil^D\right)} + 15\sqrt{D}\right) \quad (170)$$

and

$$\mathbb{E}\left[\max_{i\in[N]} G_{A_i} - \hat{Y}\right] \leq \sqrt{2}\sigma\left(\sqrt{D\log\left(\left\lceil\frac{L_k}{\log(L_k)}T^{1/D}\right\rceil\right)} - \left(1 - T^{-\frac{1}{2\sqrt{\pi}}}\right)\sqrt{\log\left(\frac{T}{3\log^{\frac{3}{2}}(T)}\right)}\right).$$

$$(171)$$

Substituting these terms in (165) yields (32).

So far, we have only shown that (32) holds when preselecting a grid as proposed in Grünewälder et al. [2010] and then restricting our optimization policies to this preselected domain. However, it is easy to show that the result also holds when allowing EI2 (Definition 2) or the UCB2 (Definition 3) to select from the entire domain $\mathcal{A}$. The proof of Corollary 1 is based on bounding the expected increment of the observed maximum at each time step (60). It is clear that by allowing the policy to select from a larger set of points, the expected increment cannot be smaller.

**Convergence**

In the following, we show that the bound converges to 0 as $T \to \infty$. Clearly, the first term in (32) converges to zero. The second term can be written (without the factor $\sqrt{2}\sigma$, as it is irrelevant) as

$$\sqrt{D \log \left( \left\lceil \frac{L_k}{\log(L_k)} T^{1/D} \right\rceil \right)} - \sqrt{\log \left( \frac{T}{3 \log^{\frac{3}{2}}(T)} \right)} + T^{-\frac{1}{2\sqrt{\pi}}} \sqrt{\log \left( \frac{T}{3 \log^{\frac{3}{2}}(T)} \right)}. \quad (172)$$

Clearly, the last of these terms converges to zero. For the other two terms we have

$$\sqrt{D \log \left( \left\lceil \frac{L_k}{\log(L_k)} T^{1/D} \right\rceil \right)} - \sqrt{\log \left( \frac{T}{3 \log^{\frac{3}{2}}(T)} \right)} \quad (173)$$

$$\leq \sqrt{D \log \left( \frac{L_k}{\log(L_k)} T^{1/D} + 1 \right)} - \sqrt{\log \left( \frac{T}{3 \log^{\frac{3}{2}}(T)} \right)} \quad (174)$$

$$= \frac{D \log \left( \frac{L_k}{\log(L_k)} T^{1/D} + 1 \right) - \log \left( \frac{T}{3 \log^{\frac{3}{2}}(T)} \right)}{\sqrt{D \log \left( \frac{L_k}{\log(L_k)} T^{1/D} + 1 \right)} + \sqrt{\log \left( \frac{T}{3 \log^{\frac{3}{2}}(T)} \right)}} \quad (175)$$

$$= \frac{D \log \left( \frac{L_k}{\log(L_k)} + T^{-1/D} \right) + \log(3) + \frac{3}{2} \log\left(\log(T)\right)}{\sqrt{D \log \left( \frac{L_k}{\log(L_k)} T^{1/D} + 1 \right)} + \sqrt{\log \left( \frac{T}{3 \log^{\frac{3}{2}}(T)} \right)}}. \quad (176)$$

The numerator grows with $\log \log T$, while the denominator grows faster, with $\sqrt{\log T}$, which means that these terms also converge to 0.

$\square$

## L   Some Properties of Expected Improvement and Related Functions

Here we prove some properties of the expected improvement ei (Definition 1) and related functions, such that we can use them in the rest of the proof.

**Lemma 9** (Convexity of expected improvement). *The standard expected improvement function* $\mathrm{ei}(x)$ *(Definition 1) is convex on* $\mathbb{R}$.

*Proof.* This is easy to see since the second derivative

$$\frac{d^2 \, \mathrm{ei}(x)}{dx^2} = \frac{e^{-\frac{x^2}{2}}}{\sqrt{2\pi}} \quad (177)$$

is positive everywhere.

$\square$

**Lemma 10** (Bounds on normal CCDF). *We have the following bounds on the complementary cumulative density function (CCDF) of the standard normal distribution*

$$\left( \tau^{-1} - \tau^{-3} \right) \mathcal{N}(\tau) \leq \Phi^c(\tau) \leq \left( \tau^{-1} - \tau^{-3} + 3\tau^{-5} \right) \mathcal{N}(\tau) \quad \forall \tau > 0. \quad (178)$$

*The normal CCDF is defined as*

$$\Phi^c(\tau) := \int_\tau^\infty \mathcal{N}(x)dx. \tag{179}$$

*Proof.* Using integration by parts, we can write for any $\tau > 0$

$$\Phi^c(\tau) = \int_\tau^\infty \mathcal{N}(x)dx \tag{180}$$

$$= \tau^{-1}\mathcal{N}(\tau) - \int_\tau^\infty x^{-2}\mathcal{N}(x)dx \tag{181}$$

$$= \left(\tau^{-1} - \tau^{-3}\right)\mathcal{N}(\tau) + 3\int_\tau^\infty x^{-4}\mathcal{N}(x)dx \tag{182}$$

$$= \left(\tau^{-1} - \tau^{-3} + 3\tau^{-5}\right)\mathcal{N}(\tau) - 15\int_\tau^\infty x^{-6}\mathcal{N}(x)dx. \tag{183}$$

From this the bounds straightforwardly follow

$\square$

**Lemma 11** (Bounds on the expected improvement)**.** *We have the following bound for the expected improvement (Definition 1)*

$$\left(\tau^{-2} - 3\tau^{-4}\right)\mathcal{N}(\tau) \le \mathrm{ei}(\tau) \le \tau^{-2}\mathcal{N}(\tau) \quad \forall \tau > 0. \tag{184}$$

*Proof.* From Definition 1 we have that

$$\mathrm{ei}(\tau) = \mathcal{N}(\tau) - \tau\Phi^c(\tau). \tag{185}$$

The desired result follows straightforwardly using the bounds from Lemma 10.

$\square$

**Lemma 12** (Upper bound on the multivariate expected improvement)**.** *For a family of jointly Gaussian distributed RVs $(F_n)_{n\in[N]}$ with mean $m \in \mathbb{R}^N$ and covariance $c \in \mathbb{S}_+^N$ and a threshold $\tau \in \mathbb{R}$, we can bound the multivariate expected improvement (Definition 8) as follows*

$$\mathrm{mei}(\tau|m,c) = \mathbb{E}\left[\max\left\{\max_{n\in[N]} F_n - \tau, 0\right\}\right] \tag{186}$$

$$\le \max\left\{\max_{n\in[N]}(m_n - \tau + \sqrt{c_{nn}2\log N}), 0\right\} + \frac{\max_{n\in[N]}\sqrt{c_{nn}}}{2\sqrt{2\pi}\log(N)}. \tag{187}$$

*Proof.* Defining $Z := \max_{n\in[N]} F_n$, we can write

$$\mathrm{mei}(0|m,c) = \mathbb{E}\left[\max\{Z,0\}\right] \tag{188}$$

$$= \int_{-\infty}^\infty \max\{z,0\}p(z)dz \tag{189}$$

$$= \int_0^\infty zp(z)dz \tag{190}$$

$$\overset{b\ge0}{=} \int_0^b zp(z)dz + \int_b^\infty zp(z)dz \tag{191}$$

$$= \int_0^b zp(z)dz + \int_0^\infty \int_b^\infty [x \le z]p(z)dzdx \tag{192}$$

$$= \int_0^b zp(z)dz + \int_0^\infty P(Z \ge \max(x,b))dx \tag{193}$$

$$= \int_0^b zp(z)dz + bP(Z \ge b) + \int_b^\infty P(Z \ge x)dx. \tag{194}$$

Since $\int_0^b zp(z)dz \leq \int_0^b bp(z)dz$ for any $b \geq 0$, we can bound this as

$$(194) \leq bP(0 \leq Z \leq b) + bP(Z \geq b) + \int_b^\infty P(Z \geq x)dx \tag{195}$$

$$= bP(0 \leq Z) + \int_b^\infty P(Z \geq x)dx \tag{196}$$

$$\leq b + \int_b^\infty P(Z \geq x)dx \tag{197}$$

$$= b + \int_b^\infty P(\vee_{n \in [N]}(F_n \geq x))dx \tag{198}$$

where we have used the fact that the event $Z \geq x$ is identical to the event $\vee_{n \in [N]}(F_n \geq x)$. Using the union bound we can now write

$$(198) \leq b + \sum_{n \in [N]} \int_b^\infty P(F_n \geq x)dx \tag{199}$$

$$= b + \sum_{n \in [N]} \int_b^\infty \int_{-\infty}^\infty [f_n \geq x]p(f_n)df_n dx \tag{200}$$

$$= b + \sum_{n \in [N]} \int_{-\infty}^\infty \max\{f_n - b, 0\}p(f_n)df_n. \tag{201}$$

Noticing that the summands in the above term match the definition of expected improvement (Definition 1), we can write

$$(201) = b + \sum_{n \in [N]} \mathrm{ei}\left(b|m_n, \sqrt{c_{nn}}\right) \tag{202}$$

$$\leq b + N \max_{n \in [N]} \mathrm{ei}\left(b|m_n, \sqrt{c_{nn}}\right) \tag{203}$$

$$= b + N \max_{n \in [N]} \sqrt{c_{nn}}\,\mathrm{ei}\left(\frac{b - m_n}{\sqrt{c_{nn}}}\right) \tag{204}$$

where we have used (11) in the last line. Using Lemma 11 we obtain for any $b > \max_n m_n$

$$(204) \leq b + N \max_{n \in [N]}\left(\sqrt{c_{nn}}\left(\frac{b - m_n}{\sqrt{c_{nn}}}\right)^{-2}\mathcal{N}\left(\frac{b - m_n}{\sqrt{c_{nn}}}\right)\right). \tag{205}$$

This inequality hence holds for any $b$ which satisfies $b > \max_n m_n$ and $b \geq 0$ (from (191)). We pick the following value which clearly satisfies these conditions

$$b = \max\left\{\max_{n \in [N]}\left(m_n + \sqrt{c_{nn}2\log N}\right), 0\right\} \tag{206}$$

from which it follows that

$$\frac{b - m_n}{\sqrt{c_{nn}}} \geq \sqrt{2\log N} \quad \forall n \in [N]. \tag{207}$$

Given this fact it is easy to see that

$$(205) \leq b + N \max_{n \in [N]}\left(\sqrt{c_{nn}}\sqrt{2\log N}^{-2}\mathcal{N}\left(\sqrt{2\log N}\right)\right) \tag{208}$$

$$= b + \frac{1}{2\sqrt{2\pi}\log N}\max_{n \in [N]}\sqrt{c_{nn}} \tag{209}$$

$$= \max\left\{\max_{n \in [N]}\left(m_n + \sqrt{c_{nn}2\log N}\right), 0\right\} + \frac{\max_{n \in [N]}\sqrt{c_{nn}}}{2\sqrt{2\pi}\log N}. \tag{210}$$

It is easy to see that $\mathrm{mei}(\tau|m, c) = \mathrm{mei}(0|m - \tau, c)$, which concludes the proof. $\qquad \square$