# OpenReview forum: "Regret Bounds for Gaussian-Process Optimization in Large Domains"
_NeurIPS.cc/2021/Conference — NeurIPS 2021 Poster_

### Official Review · Reviewer_qn7v · 2021-07-17

**Rating:** 6
**Confidence:** 4

**Summary:**

The paper studies regret bounds for Bayesian Optimization (with UCB & EI as acquisition functions) in the setting where the function domain is large relative to the number of admissible function evaluations, and the function evaluation is noiseless. The paper proposes two modified version of EI and UCB for this purpose and characterize the Bayesian simple regret of these methods, for both discrete and continuous domains. The work shows that the methods can find nontrivial function values even when the number of evaluations is far too small to find the global optimum.

**Main Review:**

Strengths:
1.	The setting studied seems novel and interesting.
2.	The paper established both upper bounds and lower bounds of their proposed algorithms.

Weaknesses:
1.	It is unclear what the exact setting the paper considers in the continuous domain and how prior work would fail in that setting (please see the Questions).
2.	Even when the paper proposes new algorithms (EI2 and UCB2) for the theoretical analysis, but it still benefits if we can see some experimental results about how BO performs with these two new algorithms. I would suggest including at least some BO experiments with the proposed algorithms EI2 and UCB2.

Questions:
1.	As mentioned, I feel unclear on the setting of the paper in the continuous domain. For discrete domain, I can understand the setting of the problem is when T (number of iterations) << N (the cardinality of the discrete domain). However, for the continuous domain, I can not find/understand the setting. I guess it has something to do with T and L_k, the Lipschitz constant of the kernel. But what exactly the setting in the continuous domain is?
2.	Lines 93-97: Why the regret bound of [Srinivas et al, 2010] is vacuous when in your setting? In other words, why $\gamma_T$ grows linearly with $T$ in your setting? I cannot follow the arguments explaining why the simple regret bound in [Srinivas et al, 2010] does not decrease at all. I think this argument needs to be proved rigorously so that you can demonstrate your proposed algorithms and analysis are necessary. I think some dedicated parts of the main paper are even needed to explain why your setting hinders [Srinivas et al, 2010].
3.	Does the bound in Theorem 2, Eq. (30) converge to 0 when T goes to infinity?  As the bound in [Grunewalder et al, 2010], Eq. (27) does converge to 0. The first term in Eq. (30) does converge to 0, but it is not trivial to derive that the 2nd term in Eq. (30) also converges to 0. Can the authors prove this?

Note: I'm willing to increase my score if the authors can address my questions properly.


**Time Spent Reviewing:**

5.5

---

> ### Author Response · Authors · 2021-08-10
> **Response**
>
> Thank you for taking the time to read the paper and giving helpful feedback! We will take it account for a potential final version of the paper. In particular, we will add an experimental evaluation of EI2/UCB2 in comparison to the standard methods EI/UCB. In addition, we will add discussions to clarify the questions raised.
>
> We address the main points of the review in individual comments, to facilitate discussion.

---

> > ### Comment · Reviewer_qn7v · 2021-08-21
> > **Responses to the authors' rebuttal**
> >
> > Hi authors,
> >
> > I'm happy with your responses regarding the linear growth of $\gamma_T$ and the setting of the work in the continuous domain. The responses make sense to me and enable me to understand and appreciate the work more.
> >
> > I decided to increase my score to 6. This is because, at the current stage, there are no experiments to evaluate the performance of EI2/UCB2 so I can not increase my score higher.

---

> ### Author Response · Authors · 2021-08-10
> **Clarifications of the continuous setting**
>
>
> This is a good point, it may indeed be helpful to explain the setting
> more explicitly. We intend to add something along the lines of the
> following discussion to the paper:
>
> In the continuous setting, we characterize the GP by $L_{k}$ and
> $\sigma$, which are properties of the kernel $k$:
> \begin{equation}
> |k(x,x)-k(x,y)|\le L_{k}\left\Vert x-y\right\Vert _{\infty}\quad\forall x,y\in\mathcal{A}
> \end{equation}
>
> \begin{align}
> k(x,x) & \le\sigma^{2}\quad\forall x\in\mathcal{A}
> \end{align}
>
> where $\mathcal{A}$ is the $D$-dimensional unit cube (note that
> to use a domain other than the unit cube we can simply rescale). The
> setting we are interested in is
> \begin{equation}
> T\ll\left(\frac{L_{k}}{2\sigma^{2}}\right)^{D}=:m(L_{k},\sigma,D)
> \end{equation}
> which implies that a large part of the GP may remain unexplored, as
> will become clear in the following comparison to related work:
>
> (Srinivas et al.) only applies when we can reduce the maximum variance
> of the GP using $T$ evaluations (see the comment on $\gamma_{T}$).
> This would require that we can acquire information on each point $x$
> that has maximum prior variance $k(x,x)=\text{max}_{z}k(z,z)=\sigma^{2}$.
> In order to ensure that we gather nonzero information on such a point
> $x$, we have to make sure to evaluate at least one point $y$ such
> that $k(x,y)>0$ (or, more realistically, $k(x,y)>\epsilon$, which
> would lead to a qualitatively similar result), which is equivalent
> to the condition
> \begin{equation}
> |k(x,x)-k(x,y)|<k(x,x)=\sigma^{2},
> \end{equation}
>
> which we can ensure by
> \begin{equation}
> L_{k}\left\Vert x-y\right\Vert _{\infty}<\sigma^{2}
> \end{equation}
>
> or equivalently by
> \begin{equation}
> \left\Vert x-y\right\Vert _{\infty}<\frac{\sigma^{2}}{L_k}.
> \end{equation}
>
> This statement says that $x$ has to be within a cube centered at
> $y$ with sidelength $2\sigma^{2}/L_{k}$. To ensure that this holds
> for all $x\in\mathcal{A}$ (since in the worst case they all have
> prior variance $\sigma^{2}$, which is typical), we need to cover
> the domain with
> \begin{equation}
> T>\left(\frac{L_{k}}{2\sigma^{2}}\right)^{D}=m(L_{k},\sigma,D)
> \end{equation}
> cubes and hence evaluations.
>
> For illustration, consider for instance the case of $L_{k}=10^{5},\sigma=1,D=2,T=10^{5}$,
> where the number of evaluations is far too low to explore the GP ($T=10^{5}\ll m(L_{k},\sigma,D)=2.5\times10^{9}$).
> Nevertheless, we see from figure 3a that our regret bounds (second-darkest
> cyan) remain low while the ones from Gruenewaelder (second-darkest
> magenta) explode in this case.
>
> (Srinivas et al.) Srinivas, N., Krause, A., Kakade, S. M., \& Seeger,
> M. (2009). Gaussian Process Optimization in the Bandit Setting: No
> Regret and Experimental Design. arXiv. http://arxiv.org/abs/0912.3995

---

> > ### Author Response · Authors · 2021-08-12
> > **Updated response**
> >
> > Note that we just updated this response, since we realized that the original response may not have properly addressed the question.

---

> ### Author Response · Authors · 2021-08-10
> **Linear growth of $\gamma_T$**
>
> Thanks for pointing out the need to be more formal on this point:
> From Lemma 5.3 in (Srinivas et al.) (see the reference below, we use
> slightly different notation) we have that the information gain of
> a set of points $X=\\{x_1,...,x_T\\}$ can be expressed as
>
> \begin{equation}
> G(X):=I(y_{X};f_{X})=\frac{1}{2}\sum_{t=1}^{T}\log(1+\sigma_{y}^{-2}\sigma^{2}(x_{t}|x_{1:t-1}))
> \end{equation}
>
> where $\sigma^{2}(x_{t}|x_{1:t-1})$ is the predictive variance after
> evaluating at $x_{1:t-1}$ and $\sigma_{y}^{2}$ is the variance of
> the observation noise$^{*}$. Hence, we can write the information
> gain for $T+1$ points as
>
> \begin{equation}
> G(X\cup\{x_{T+1}\})=G(X)+\frac{1}{2}\log(1+\sigma_{y}^{-2}\sigma^{2}(x_{T+1}|x_{1:T})).
> \end{equation}
>
> Now let $X^{*}:=\max_{X:|X|=T}G(X)$ be the points that maximize the
> information gain. By definition (see equation 7 in (Srinivas et al.)),
> we have
>
> \begin{align}
> \gamma_{T} & :=G(X^{*})
> \end{align}
>
> that is, $\gamma_{T}$ is the maximum information that can be acquired
> using $T$ points. For $T+1$ points we have
>
> \begin{align}
> \gamma_{T+1} & =\max_{X,x_{T+1}}G(X\cup\{x_{T+1}\})
> \end{align}
>
> \begin{align}
>  & \ge\max_{x_{T+1}}G(X^{*}\cup\{x_{T+1}\})
> \end{align}
>
> where the inequality follows from the fact that maximizing over $X,x_{T+1}$
> jointly will at least yield as high a value as just picking $X^{*}$
> from the previous optimization and optimizing only over $x_{T+1}$.
> Plugging in $G(X\cup\{x_{T+1}\})$ from above, we have
>
> \begin{equation}
> \gamma_{T+1}\ge G(X^*)+\frac{1}{2}\max_{x_{T+1}}\log(1+\sigma_{y}^{-2}\sigma^{2}(x_{T+1}|x_{1:T}^{*}))
> \end{equation}
>
> and hence
> \begin{equation}
> \gamma_{T+1}\ge\gamma_{T}+\frac{1}{2}\max_{x_{T+1}}\log(1+\sigma_{y}^{-2}\sigma^{2}(x_{T+1}|x_{1:T}^{*})).
> \end{equation}
>
> This means that if $T$ is not large enough to explore the GP reasonably
> well everywhere (i.e., there are still $x$ such that $\sigma^{2}(x|x_{1:T}^{*})$
> is large), then adding an additional observation can add substantial
> information, i.e. $\gamma_{T+1}$ is substantially larger than $\gamma_{T}$
> (which means the regret grows substantially).
>
> As a more concrete case, suppose we have a GP which a priori has a
> uniform variance $\sigma^{2}(x)=s^{2}\quad\forall x$. In addition,
> suppose that the GP domain is large with respect to $T$, in the sense
> that it is not possible to reduce the variance everywhere substantially
> by observing $T$ (or less) points, i.e. we have $\max_{x_{t}}\sigma(x_{t}|x_{1:t-1})\approx s~\forall x_{1:t-1},t\le T+1$.
> We hence have
>
> \begin{equation}
> \gamma_{T}=\max_{x_{1:T}}\frac{1}{2}\sum_{t=1}^{T}\log(1+\sigma_{y}^{-2}\sigma^{2}(x_{t}|x_{1:t-1}))
> \end{equation}
>
> \begin{equation}
> \approx\frac{1}{2}T\log(1+\sigma_{y}^{-2}s^{2}).
> \end{equation}
>
> This linear growth will continue until $T$ is large enough such that
> the uncertainty of the GP can be reduced substantially everywhere.
>
> Since the bound on the cumulative regret $R_{T}$ is of the form $\sqrt{T\gamma_{T}}$
> (see Theorem 1 in (Srinivas et al.)) it will hence also grow linearly
> in $T$. (Srinivas et al.) then bound the suboptimality of the optimization
> by the average regret $R_{T}/T$ (see the paragraph on regret in section
> 2 of (Srinivas et al.)), which however does not decrease as long as
> $R_{T}$ grows linearly in $T$.
>
> $^{*}$Note that the information gain goes to infinity as $\sigma_y$
> goes to zero, which is in fact another reason why the results from
> (Srinivas et al.) are not directly applicable to our setting. We did
> not discuss this point in the paper because it is a technicality that
> can be resolved (in the most naive way, one could add artificial noise).
>
> (Srinivas et al.) Srinivas, N., Krause, A., Kakade, S. M., \& Seeger,
> M. (2009). Gaussian Process Optimization in the Bandit Setting: No
> Regret and Experimental Design. arXiv. http://arxiv.org/abs/0912.3995

---

> ### Author Response · Authors · 2021-08-10
> **Convergence of the continuous bound**
>
>
> This is an interesting question and we will include the following
> discussion into a potential final version of the paper:
>
> Yes, the bound in Theorem 2 does indeed converge to 0. We can write
> the second term of equation 30 (without the factor $\sqrt{2}\sigma$,
> as it is irrelevant) as
>
> \begin{equation}
> \sqrt{D\log\left(\left\lceil \frac{L_{k}}{\log(L_{k})}T^{1/D}\right\rceil \right)}-\sqrt{\log\left(\frac{T}{3\log^{\frac{3}{2}}(T)}\right)}+T^{-\frac{1}{2\sqrt{\pi}}}\sqrt{\log\left(\frac{T}{3\log^{\frac{3}{2}}(T)}\right)}.
> \end{equation}
> Clearly, the last term converges to zero. For the other two terms
> we have
> \begin{equation}
> \sqrt{D\log\left(\left\lceil \frac{L_{k}}{\log(L_{k})}T^{1/D}\right\rceil \right)}-\sqrt{\log\left(\frac{T}{3\log^{\frac{3}{2}}(T)}\right)}
> \end{equation}
> \begin{equation}
> \le\sqrt{D\log\left(\frac{L_{k}}{\log(L_{k})}T^{1/D}+1\right)}-\sqrt{\log\left(\frac{T}{3\log^{\frac{3}{2}}(T)}\right)}
> \end{equation}
> \begin{equation}
> =\frac{D\log\left(\frac{L_{k}}{\log(L_{k})}T^{1/D}+1\right)-\log\left(\frac{T}{3\log^{\frac{3}{2}}(T)}\right)}{\sqrt{D\log\left(\frac{L_{k}}{\log(L_{k})}T^{1/D}+1\right)}+\sqrt{\log\left(\frac{T}{3\log^{\frac{3}{2}}(T)}\right)}}
> \end{equation}
> \begin{equation}
> =\frac{D\log\left(\frac{L_{k}}{\log(L_{k})}+T^{-1/D}\right)+\log\left(3\right)+\frac{3}{2}\log\left(\log(T)\right)}{\sqrt{D\log\left(\frac{L_{k}}{\log(L_{k})}T^{1/D}+1\right)}+\sqrt{\log\left(\frac{T}{3\log^{\frac{3}{2}}(T)}\right)}}.
> \end{equation}
> The numerator grows with $\log\log T$, while the denominator grows
> faster, with $\sqrt{\log T}$, which means that these terms also converge
> to 0.

---

### Official Review · Reviewer_sZcy · 2021-07-17

**Rating:** 7
**Confidence:** 3

**Summary:**

This paper studies the problem of Gaussian process optimization when the number of arms (the domain) is larger than the horizon of the problem. In the finite arm case, it is assumed that the function is drawn from a zero-mean Gaussian prior and the learner is allowed to make T noiseless queries. Two algorithms based on expected improvement and UCB are proposed to solve this by achieving a simple regret that is some fraction of the optimal value. These are shown to be tight. It is shown that similar results can be extended to the continuous domain, but instead depending on a large Lipschitz constant.

**Limitations And Societal Impact:**

Limitations and suggestions discussed in the main review.

**Main Review:**

Overall, I think this is a good paper with a solid and reasonably thorough contribution. The problem setting is very interesting where one must make do with potentially very little data compared to the complexity of the problem. I think this class of problems is of great relevance for the bandit/GP/online learning communities. The related work seems to do a good job of placing the paper in the current literature. The paper itself is well-written and easy to understand.

The limitations of the paper are already discussed in some detail in the paper’s limitations section. To reiterate a couple that I agree are particularly important: (1) the algorithms seem to be non-standard, aiming to reduce uncertainty in both the best and worst estimates. (2) The observations are noiseless.

Regarding (1), in the more traditional setting when many evaluations are actually allowed, do the proposed algorithms achieve comparable performance with the state-of-the-art? I think this would be useful in understanding what trade-offs (if any) are necessary to achieve good performance in the large domain regime. Similarly, is there simple intuition for what makes the analysis of the standard algorithms fail in the large domain regime? It was mentioned in the paper that this was the case, but it was unclear to me why. It seems like whether there is a separation between these two regimes is an open question.

Regarding (2), this seems like a particularly limiting aspect of the results. It is also difficult for me to tell whether handling noise in the observations would require a non-trivial extension of current analysis. Can you elaborate on this? If the extension would be trivial, I would suggest including it perhaps in the appendix in order to make the work more complete.


**Time Spent Reviewing:**

4

---

> ### Author Response · Authors · 2021-08-10
> **Response**
>
> Thank you for taking the time to read the paper and giving helpful feedback! We will take it account for a potential final version of the paper.
>
> In the following we will address the main points of the review in individual comments, to facilitate discussion.

---

> ### Author Response · Authors · 2021-08-10
> **Do EI2/UCB2 achieve state-of-the-art performance in the traditional setting?**
>
> We expect that EI2/UCB2 perform comparably to standard EI/UCB in the
> in the traditional setting (up to a factor 2 in the number of evaluations),
> where many evaluations are allowed, for the following reasons:
>
> For a centered GP, by symmetry, EI2/UCB2 will in expectation perform
> as many maximum evaluations as minimum evaluations. Therefore, we
> would expect that EI2/UCB2 with $2T$ evaluations performs at least
> as well as EI/UCB with $T$ evaluations. In both cases the number
> of maximizations is $T$, and in the case if EI2/UCB2 we have in addition
> $T$ minimizations. These minimizations merely add some additional
> information about the GP, which intuitively should not worsen performance.
>
> Formally proving this in general seems nontrivial, but we can show
> quite easily that the regret bounds from (Srinivas et al.) hold. In
> particular, Theorem 1 from (Srinivas et al.) holds if we run UCB2
> (the symmetrized version of their UCB) until it performed $T$ maximizations,
> with $R_{T}$ being the cumulative regret of the maximizations only
> (discarding the minimizations for the regret computation is unproblematic
> here, since we only care about the maximal value obtained). Theorem
> 1 holds because the maximum information gain $\gamma_{T}$ that can
> be obtained through the $T$ maximizations cannot increase through
> intermediate minimizations (since they cannot increase the entropy
> of the GP).
>
> (Srinivas et al.) Srinivas, N., Krause, A., Kakade, S. M., \& Seeger,
> M. (2009). Gaussian Process Optimization in the Bandit Setting: No
> Regret and Experimental Design. arXiv. http://arxiv.org/abs/0912.3995

---

> ### Author Response · Authors · 2021-08-10
> **What makes the analysis of standard EI/UCB fail in our setting?**
>
> We use a quite different analysis technique (based on adaptive submodularity)
> compared to related work. In particular, for our proofs the symmetry
> of EI2/UCB2 is helpful (see appendix B.4): Whenever $\hat{\check{Y}}$is
> substantially smaller than $\hat{\check{F}}$, we show that it is
> possible to either increase $\hat{Y}$ or decrease $\check{Y}$ substantially.
>
> Nevertheless, we are able to give regret bounds for standard EI/UCB,
> we did not include them in the paper because they are weaker than
> the ones given in the paper. However, we now believe that it is informative
> to include them into a final version of the paper. In this case, instead
> of $\hat{\check{Y}}=\hat{Y}-\check{Y}$ we give guarantees on $\hat{Y}-\check{F}$,
> i.e. how much larger the best observation is than the function minimum.
> The difficulty here is that $\check{F}$ is not a quantity that can
> easily be expressed analytically (in contrast to the case of EI2/UCB2,
> where we used $\check{Y}$ instead, which is an observable). Therefore, we had to use a bound
> on $\check{F}$ in this proof (we are happy to go provide more details,
> if desired), which leads to weaker regret bounds than the ones presented
> in the paper.
>
>
>
> We obtain the following version of Theorem 1 for standard EI/UCB:
>
> For any problem instance $(N,T,\mu,\Sigma)$ with $N\ge T\ge500$,
> if we follow either the $\text{\textbf{standard EI or UCB strategy}}$
> we have
> \begin{equation}
> \frac{\mathbb{E}\left[\hat{\check{F}}\right]-\mathbb{E}\left[\boldsymbol{\hat{Y}-\check{F}}\right]}{\mathbb{E}\left[\hat{\check{F}}\right]}\le1-\left(1-T^{-\frac{1}{2\sqrt{\pi}}}\right)\sqrt{\frac{\log(T)-\log\left(3\log^{\frac{3}{2}}(T)\right)}{\log(N)}}.
> \end{equation}
> Note that we marked the changes in bold. Now, instead of a guarantee
> on $\hat{\check{Y}}$, we provide a guarantee on $\hat{Y}-\check{F}$,
> i.e. the difference between the best obtained value and the function
> minimum. We can then derive from that a version of Corollary 1:
>
> For any problem instance $(N,T,\mu,\Sigma)$ with zero mean $\mu_{n}=0~\forall n$
> and $N\ge T\ge500$, if we follow either the $\text{\textbf{standard EI or UCB strategy}}$,
> we have
> \begin{equation}
> normreg\le\boldsymbol{2}\left(1-\left(1-T^{-\frac{1}{2\sqrt{\pi}}}\right)\sqrt{\frac{\log(T)-\log\left(3\log^{\frac{3}{2}}(T)\right)}{\log(N)}}\right).
> \end{equation}
> The important thing to note here is the appearance of the factor 2
> in the bound. This means that asymptotically we have
> \begin{equation}
> normreg\le\boldsymbol{2}\left(1-\frac{\sqrt{\log T}}{\sqrt{\log N}}\right)
> \end{equation}
> and
> \begin{equation}
> T\le N^{(1-normreg/\boldsymbol{2})}
> \end{equation}
> which is weaker compared to equations (21),(22). Nevertheless, this
> result sheds some light on the applicability of the analysis in this
> paper to standard EI/UCB and we will include it in a potential final
> version of the paper.

---

> ### Author Response · Authors · 2021-08-10
> **Observation noise**
>
> Our goal here was to focus on the issue of large domains, without
> the added difficulty of noisy observations, such as to allow a clearer
> view of the core problem. Nevertheless, we agree that it would be
> very interesting to understand how the two problems interact and we
> will add a discussion of this point to a potential final version of
> the paper.
>
> Interestingly, the proofs apply practically without any changes to
> the setting with observation noise. The caveat is that the regret
> bounds are on the largest $\text{\textbf{noisy observation}}$ $\hat{Y}$
> rather than the largest retrieved $\text{\textbf{function value}}$
> $\max_{t}F_{A_{t}}$ (the two are identical in the noise-free setting).
>
> As a naive way of obtaining regret bounds on $\max_{t}F_{A_{t}}$,
> one could simply evaluate each point $n$ times and use the average
> observation as a pseudo observation. Choosing $n$ large enough, all
> pseudo observations $Y_{1:T}$ will be close to their respective function
> values $F_{A_{1:T}}$ with high probability. To guarantee that all
> $T$ pseudo observations are within $\epsilon$ of the true function
> values with probability $\delta$, we would need $n=\log(T/\delta)f(\sigma_{y},\epsilon)$
> (this follows from union bound over $T$ observations), where $f$
> is some function that is not relevant here and $\sigma_{y}$ is the
> noise standard deviation. We can now simply replace $T$ with $T/\left(\log(T/\delta)f(\sigma_{y},\epsilon)\right)$
> in all the theorems (to be precise, we would also have to add $\epsilon$
> to the regret, but it can be made arbitrarily small).
>
> While this solution is impractical, it is interesting to note that
> the dependence of the resulting regret-bounds on the domain size $N$
> and Lipschitz constant $L_{k}$ does not change. The dependence on
> $T$ is also identical, up to a $\log$ factor. This suggests that
> the relations we uncovered in this paper between the regret, the number
> of evaluations $T$, the domain size $N$, the Lipschitz constant
> $L_{k}$ remain qualitatively the same in the presence of observation
> noise.

---

### Official Review · Reviewer_i5A5 · 2021-07-22

**Rating:** 7
**Confidence:** 3

**Summary:**

This paper considers Gaussian process optimization and provides bounds for the (normalized) Bayesian simple regret of two policies. Each of these policies can be seen as a symmetrization of either the classical expected improvement (EI) or upper confidence bound (UCB) policies. The symmetrized version of EI, which is denoted by EI2,  picks the point expected to either increase the current observed maximum or decrease the current observed minimum the most. UCB2 is defined analogously with respect to UCB. The bounds obtained are shown to be tight and suggest that it is possible to find points with high (normalized) objective value even when the number of evaluations is small relative to the number of points in the domain, $N$,  (in the finite case, or relative to the smoothness of the prior covariance function in the continuous case), although it still grows polynomially with respect to $N$. Such guarantee contrasts with previously obtained results, which require the number of evaluations to be large with respect to the number of points in the domain.

**Limitations And Societal Impact:**

The authors have properly discussed the limitations of their work, particularly with respect to the practical relevance of the EI2 and UCB2 policies.

Understanding the performance of myopic Bayesian optimization algorithms in the low-budget regime is of great relevance and has the potential to shape the landscape of future research in Bayesian optimization. As a consequence, works of this nature could have a significant societal impact. I encourage the authors to discuss this in more detail.

**Main Review:**

Originality: This paper contains several ideas that are, to the best of my knowledge, novel. Perhaps the most striking one is the consideration of the symmetrized versions of the EI and UCB policies. Although these policies are likely to have no practical impact, it is interesting to see that there exist simple myopic policies guaranteed to achieve low (normalized) regret in the low-budget regime. Moreover, this proof technique is likely to inspire similar works in other GP-based active learning settings.

Quality: Most of the claims are well supported, and the main ideas in the proofs of the two main results seem to be right (although did not verify in detail most of the calculations). On the other hand, I would encourage the authors to provide a more formal treatment of the following two claims: (1) "As long as the GP still has large uncertainty... $\gamma_T$ will essentially
grow linearly in $T$"; (2) "the problem at hand is not adaptively submodular". While I understand that the main results proved in this paper do not rely on these two claims, a more formal treatment would help to clarify the technical contributions made with respect to previous works.

Clarity: This paper is very well written and, most of the time, has a nice balance between intuitive explanations and formal statements. The mathematical notation is well chosen. Figures 1 and 2 help understanding the significance of the obtained bounds. On the other hand, I would encourage the authors to discuss how the bounds obtained in the finite setting relate to those in the continuous setting. The relationship between these two results can prove somewhat mysterious for a reader omitting the proof of Theorem 2.

Significance: This paper provides an interesting approach to understanding the performance of simple myopic Bayesian optimization algorithms in the low-budget regime. Although the policies considered here are likely to have no practical impact, it is interesting to see that there exist simple myopic policies guaranteed to achieve low (normalized) regret in the low-budget regime.  At the same time, it would be nice if the performance of the EI2 and UCB2 policies could be related to that of the classical EI and UCB policies,  even at the expense of proving weaker bounds. Finally, I consider that an empirical assessment of the EI2 and UCB2 policies is desirable even if they underperform their classical counterparts.

Other comments:

The following paper should be discussed as part of the related work: Ryzhov, I. O. (2016). On the convergence rates of expected improvement methods. Operations Research, 64(6), 1515-1528.

**Time Spent Reviewing:**

6 hours

---

> ### Author Response · Authors · 2021-08-10
> **Response**
>
> Thank you for taking the time to read the paper and to give helpful feedback! We will take it account for a potential final version of the paper.
>
> In the following we will address the main points of the review in individual comments, to facilitate discussion.

---

> > ### Comment · Reviewer_i5A5 · 2021-08-22
> > **Thoughts after reading authors' response and empirical evaluation**
> >
> > Dear authors,
> >
> > Thank you for your detailed response, which has addressed most of my concerns.
> >
> > Before making my final decision about this paper, I would like to hear more details about the empirical evaluation you are planning to conduct and, if possible, see some preliminary results. I do not want to speak for my fellow reviewers but I think this is something that would be appreciated by several of us.
> >
> > My hope is that your empirical evaluation sheds some light on the conditions under which EI2 and UCB2 perform well/badly with respect to their classical counterparts.

---

> > > ### Author Response · Authors · 2021-08-24
> > > **Working on empirical evaluation**
> > >
> > > Dear reviewer,
> > >
> > > We are glad to hear that we were able to address most concerns. Regarding the empirical evaluation: we are looking into it and will get back to you this week.

---

> > > ### Author Response · Authors · 2021-08-26
> > > **Empirical evaluation**
> > >
> > > We conducted within-model experiments (where the ground-truth function
> > > is a sample drawn from the GP) similar to (Srinivas et al., 2010).
> > > We defined a GP $G$ on a $D$-dimensional unit cube with a squared-exponential
> > > kernel with length scale $l$. The smaller the length-scale and the
> > > larger the dimensionality, the harder the problem. For the experiments
> > > that follow, we chose the ranges of $D,l$ such that we cover the
> > > classical setting (where the global optimum can be identified with the available number of evaluations) as
> > > well as the large-domain setting (where this is not possible).
> > >
> > > The expected regret
> > > \begin{equation}
> > > r(l,D,T):=\mathbb{E}\left[\sup_{a\in\mathcal{A}}G(a)-\hat{Y}\right]
> > > \end{equation}
> > > is a function of the length-scale $l$, the dimension $D$ and the
> > > number of function evaluations $T$. We compute this quantity empirically
> > > using $50$ samples (i.e. we run each algorithm with $50$ randomly-drawn ground-truth functions).
> > > While (Srinivas et al., 2010) and other related papers typically plot $r$ as a function of $T$ (which we will also include in a final version of the paper),
> > > it is more instructive for us to look at how many evaluations $T$
> > > are required to attain a given regret $R=r(l,D,T)$:
> > > \begin{equation}
> > > T=t(R,l,D).
> > > \end{equation}
> > > We can then compare the required number of steps for EI and EI2 using
> > > \begin{equation}
> > > \frac{t_{ei}(R,l,D)}{t_{ei2}(R,l,D)},
> > > \end{equation}
> > > which we report in the following table:
> > >
> > >
> > >
> > > | ($D, l$)           |    R=0.8 |    R=0.4 |    R=0.1 |   R=0.01 |
> > > |:-----------|-------:|-------:|-------:|-------:|
> > > | (1, 0.003) |   0.87 | nan    | nan    | nan    |
> > > | (1, 0.03)  |   0.83 |   0.91 |   0.76 |   0.77 |
> > > | (1, 0.3)   |   1    |   1    |   1    |   0.8  |
> > > | (2, 0.003) | nan    | nan    | nan    | nan    |
> > > | (2, 0.03)  | nan    | nan    | nan    | nan    |
> > > | (2, 0.3)   |   1    |   0.57 |   0.82 |   0.78 |
> > > | (4, 0.003) | nan    | nan    | nan    | nan    |
> > > | (4, 0.03)  | nan    | nan    | nan    | nan    |
> > > | (4, 0.3)   |   0.75 |   0.82 | nan    | nan    |
> > >
> > >
> > >
> > > The nan entries correspond to the case where our simulation did not
> > > run for enough evaluations to attain the given regret (we will conduct
> > > this experiment with a longer time-horizon for a potential final version
> > > of the paper). From the other entries, we can see that $0.5t_{ei2}\le t_{ei}\le t_{ei2}$,
> > > which means that EI always reaches the given expected regret $R$
> > > faster than EI2, but not more than twice as fast. This is what we
> > > intuitively expected: EI should do better than EI2, because it does
> > > not waste evaluations on minimization, but we would expect it to use at least half as many evaluations as EI2, since in expectation every second
> > > evaluation of EI2 is a maximization. Note that the entries which are
> > > $1$, i.e. both algorithms perform equally well, correspond to particularly
> > > simple settings (large $l$, low $D$) where both algorithms find
> > > solutions with regret $R$ in just a handful of evaluations.
> > >
> > > For UCB and UCB2 (we use the same, fixed confidence level for both) we obtain
> > > similar results:
> > >
> > >
> > > | ($D, l$)           |    R=0.8 |    R=0.4 |    R=0.1 |   R=0.01 |
> > > |:-----------|-------:|-------:|-------:|-------:|
> > > | (1, 0.003) |   0.87 | nan    | nan    | nan    |
> > > | (1, 0.03)  |   0.86 |   0.85 |   0.87 |   0.81 |
> > > | (1, 0.3)   |   1    |   1    |   1    |   1    |
> > > | (2, 0.003) | nan    | nan    | nan    | nan    |
> > > | (2, 0.03)  |   0.84 | nan    | nan    | nan    |
> > > | (2, 0.3)   |   1    |   0.86 |   0.77 |   0.76 |
> > > | (4, 0.003) | nan    | nan    | nan    | nan    |
> > > | (4, 0.03)  | nan    | nan    | nan    | nan    |
> > > | (4, 0.3)   |   0.79 |   0.66 | nan    | nan    |
> > >
> > > For a final version of the paper, we will conduct experiments with a longer time-horizon (which we did not do yet due to computational time), such that more entries in the table can be filled, and we will consider a wider range of dimensions and length-scales. In addition, we will plot the expected regret $r$ as a function of the number of evaluations $T$ for EI/EI2 and UCB/UCB2.
> > > Please let us know if you think any other experiments would be important.

---

> > > > ### Comment · Reviewer_i5A5 · 2021-09-04
> > > > **Final thoughts on this paper**
> > > >
> > > > Dear authors,
> > > >
> > > > Thank you for presenting the preliminary results of your empirical evaluation. I look forward to seeing the complete analysis in the revised version of your paper. I wonder if it would also be educational to presents experiments in the finite domain setting and shed light on how $\Sigma$ affects the performance of EI2 and UCB2 relative to their classical counterparts. I am guessing that the performance gap will be larger as the correlation between the arms increases. Is there a difference between higher positive and negative correlation?  What other factors could result in a larger performance gap? I think these are interesting questions and would be interested in hearing your  thoughts on this (even if no results are presented at the moment)
> > > >
> > > > The contribution of this paper is significant and, while it still requires some work before it is ready for publication, I believe it is realistic to be completed on time in case of being accepted.  I have thus decided to increase my score slightly.
> > > >
> > > > Best wishes,
> > > >
> > > > Reviewer i5A5

---

> > > > > ### Author Response · Authors · 2021-09-07
> > > > > **Empirical evaluation in the finite-domain setting**
> > > > >
> > > > > Dear reviewer,
> > > > >
> > > > > Thank you for your insightful comments! Conducting such experiments in the finite-domain setting would indeed be very interesting, since there are no constraints on the covariance matrix in this setting. We would still expect that EI2/UCB2 would require no more than 2x the number of evaluations that EI/UCB use, since also in this setting EI2/UCB2 are expected to perform half of their evaluations as maximizations.
> > > > >
> > > > > We agree with your intuition that the performance gap may grow with the correlation between arms, since evaluating at potential maxima will typically yield information about other potential maxima, in the case of positively correlated arms. The case of negatively-correlated arms is far less intuitive. A curious toy example is $F_{1:n/2}=-F_{n/2+1,n}$, i.e. the function contains a negated copy of itself. In that case, by symmetry, EI2/UCB2 will perform identically to EI/UCB.
> > > > >
> > > > > We will add an empirical assessment of these questions. We plan to use band covariance matrices, where we vary the off-diagonal elements from positive to negative (in a range that ensures positive-definiteness). We also plan to conduct evaluations using randomly-generated covariance matrices, such as to gain an intuition how the performance gap may depend on the covariance matrix.
> > > > >
> > > > >
> > > > > Best wishes,
> > > > >
> > > > > the authors

---

> ### Author Response · Authors · 2021-08-10
> **Linear growth of $\gamma_T$**
>
> Thanks for pointing out the need to be more formal on this point:
> From Lemma 5.3 in (Srinivas et al.) (see the reference below, we use
> slightly different notation) we have that the information gain of
> a set of points $X=\\{x_1,...,x_T\\}$ can be expressed as
>
> \begin{equation}
> G(X):=I(y_{X};f_{X})=\frac{1}{2}\sum_{t=1}^{T}\log(1+\sigma_{y}^{-2}\sigma^{2}(x_{t}|x_{1:t-1}))
> \end{equation}
>
> where $\sigma^{2}(x_{t}|x_{1:t-1})$ is the predictive variance after
> evaluating at $x_{1:t-1}$ and $\sigma_{y}^{2}$ is the variance of
> the observation noise$^{*}$. Hence, we can write the information
> gain for $T+1$ points as
>
> \begin{equation}
> G(X\cup\{x_{T+1}\})=G(X)+\frac{1}{2}\log(1+\sigma_{y}^{-2}\sigma^{2}(x_{T+1}|x_{1:T})).
> \end{equation}
>
> Now let $X^{*}:=\max_{X:|X|=T}G(X)$ be the points that maximize the
> information gain. By definition (see equation 7 in (Srinivas et al.)),
> we have
>
> \begin{align}
> \gamma_{T} & :=G(X^{*})
> \end{align}
>
> that is, $\gamma_{T}$ is the maximum information that can be acquired
> using $T$ points. For $T+1$ points we have
>
> \begin{align}
> \gamma_{T+1} & =\max_{X,x_{T+1}}G(X\cup\{x_{T+1}\})
> \end{align}
>
> \begin{align}
>  & \ge\max_{x_{T+1}}G(X^{*}\cup\{x_{T+1}\})
> \end{align}
>
> where the inequality follows from the fact that maximizing over $X,x_{T+1}$
> jointly will at least yield as high a value as just picking $X^{*}$
> from the previous optimization and optimizing only over $x_{T+1}$.
> Plugging in $G(X\cup\{x_{T+1}\})$ from above, we have
>
> \begin{equation}
> \gamma_{T+1}\ge G(X^*)+\frac{1}{2}\max_{x_{T+1}}\log(1+\sigma_{y}^{-2}\sigma^{2}(x_{T+1}|x_{1:T}^{*}))
> \end{equation}
>
> and hence
> \begin{equation}
> \gamma_{T+1}\ge\gamma_{T}+\frac{1}{2}\max_{x_{T+1}}\log(1+\sigma_{y}^{-2}\sigma^{2}(x_{T+1}|x_{1:T}^{*})).
> \end{equation}
>
> This means that if $T$ is not large enough to explore the GP reasonably
> well everywhere (i.e., there are still $x$ such that $\sigma^{2}(x|x_{1:T}^{*})$
> is large), then adding an additional observation can add substantial
> information, i.e. $\gamma_{T+1}$ is substantially larger than $\gamma_{T}$
> (which means the regret grows substantially).
>
> As a more concrete case, suppose we have a GP which a priori has a
> uniform variance $\sigma^{2}(x)=s^{2}\quad\forall x$. In addition,
> suppose that the GP domain is large with respect to $T$, in the sense
> that it is not possible to reduce the variance everywhere substantially
> by observing $T$ (or less) points, i.e. we have $\max_{x_{t}}\sigma(x_{t}|x_{1:t-1})\approx s~\forall x_{1:t-1},t\le T+1$.
> We hence have
>
> \begin{equation}
> \gamma_{T}=\max_{x_{1:T}}\frac{1}{2}\sum_{t=1}^{T}\log(1+\sigma_{y}^{-2}\sigma^{2}(x_{t}|x_{1:t-1}))
> \end{equation}
>
> \begin{equation}
> \approx\frac{1}{2}T\log(1+\sigma_{y}^{-2}s^{2}).
> \end{equation}
>
> This linear growth will continue until $T$ is large enough such that
> the uncertainty of the GP can be reduced substantially everywhere.
>
> Since the bound on the cumulative regret $R_{T}$ is of the form $\sqrt{T\gamma_{T}}$
> (see Theorem 1 in (Srinivas et al.)) it will hence also grow linearly
> in $T$. (Srinivas et al.) then bound the suboptimality of the optimization
> by the average regret $R_{T}/T$ (see the paragraph on regret in section
> 2 of (Srinivas et al.)), which however does not decrease as long as
> $R_{T}$ grows linearly in $T$.
>
> $^{*}$Note that the information gain goes to infinity as $\sigma_y$
> goes to zero, which is in fact another reason why the results from
> (Srinivas et al.) are not directly applicable to our setting. We did
> not discuss this point in the paper because it is a technicality that
> can be resolved (in the most naive way, one could add artificial noise).
>
> (Srinivas et al.) Srinivas, N., Krause, A., Kakade, S. M., \& Seeger,
> M. (2009). Gaussian Process Optimization in the Bandit Setting: No
> Regret and Experimental Design. arXiv. http://arxiv.org/abs/0912.3995

---

> ### Author Response · Authors · 2021-08-10
> **Why is the problem not adaptively submodular?**
>
> We discuss this point in more detail in appendix A.1. In a final version of this paper, we will make sure to clarify this point in the main text and refer to appendix A.1. Please let us know in case this does not answer the question.

---

> ### Author Response · Authors · 2021-08-10
> **Relation between finite and continuous setting**
>
> We will add such a discussion to a final version of the paper. We plan to expand the discussion of lines 270-272 and 279-282 and move them to the beginning of the section, such that it is immediately clear what the path from the finite setting to the continuous setting is.

---

> ### Author Response · Authors · 2021-08-10
> **Relation to standard EI/UCB**
>
> Intuitively, we would expect standard EI/UCB to perform better than
> EI2/UCB2 for maximization: At any given step, evaluating at a potential
> maximizer instead of a minimizer will clearly lead to a larger immediate
> reduction in regret (i.e. for the last optimization step, it would
> be clearly the better choice). However, what is hard to quantify is
> how the information acquired by this evaluation will influence the
> future behavior of the algorithm. Intuitively, we would expect that
> an evaluation at a potential minimizer will not provide any more useful
> information than an evaluation at a potential maximizer. If we could
> prove this to be the case, this would imply that EI/UCB perform better
> than EI2/UCB2 for maximization, and hence that our bounds apply to
> EI/UCB. Unfortunately, proving this formally appears to be surprisingly
> difficult.
>
> Nevertheless, we are able to give regret bounds for standard EI/UCB
> with some small modifications to the proof (we are happy to go into
> more detail, if desired). We did not include these bounds in the paper
> because they are weaker than the ones we derived for EI2/UCB2, but
> we discuss them in the following and intend to include them into a
> final version of the paper. In addition, we will add an empirical
> comparison between EI/UCB and EI2/UCB2.
>
>
>
> We obtain the following version of Theorem 1 for standard EI/UCB:
>
> For any problem instance $(N,T,\mu,\Sigma)$ with $N\ge T\ge500$,
> if we follow either the $\text{\textbf{standard EI or UCB strategy}}$
> we have
> \begin{equation}
> \frac{\mathbb{E}\left[\hat{\check{F}}\right]-\mathbb{E}\left[\boldsymbol{\hat{Y}-\check{F}}\right]}{\mathbb{E}\left[\hat{\check{F}}\right]}\le1-\left(1-T^{-\frac{1}{2\sqrt{\pi}}}\right)\sqrt{\frac{\log(T)-\log\left(3\log^{\frac{3}{2}}(T)\right)}{\log(N)}}.
> \end{equation}
> Note that we marked the changes in bold. Now, instead of a guarantee
> on $\hat{\check{Y}}$, we provide a guarantee on $\hat{Y}-\check{F}$,
> i.e. the difference between the best obtained value and the function
> minimum. We can then derive from that a version of Corollary 1:
>
> For any problem instance $(N,T,\mu,\Sigma)$ with zero mean $\mu_{n}=0~\forall n$
> and $N\ge T\ge500$, if we follow either the $\text{\textbf{standard EI or UCB strategy}}$,
> we have
> \begin{equation}
> normreg\le\boldsymbol{2}\left(1-\left(1-T^{-\frac{1}{2\sqrt{\pi}}}\right)\sqrt{\frac{\log(T)-\log\left(3\log^{\frac{3}{2}}(T)\right)}{\log(N)}}\right).
> \end{equation}
> The important thing to note here is the appearance of the factor 2
> in the bound. This means that asymptotically we have
> \begin{equation}
> normreg\le\boldsymbol{2}\left(1-\frac{\sqrt{\log T}}{\sqrt{\log N}}\right)
> \end{equation}
> and
> \begin{equation}
> T\le N^{(1-normreg/\boldsymbol{2})}
> \end{equation}
> which is weaker compared to equations (21),(22). Nevertheless, this
> result sheds some light on the applicability of the analysis in this
> paper to standard EI/UCB and we will include it in a potential final
> version of the paper.

---

> ### Author Response · Authors · 2021-08-10
> **Related work and societal impact**
>
> We will make sure to discuss the work by Ryzhov, I. O. (2016) in the related work section. They present very interesting asymptotic results that are complementary to ours.
>
>
> We will also discuss the societal impact in more detail in a final version of the paper.

---

### Official Review · Reviewer_PkGh · 2021-07-28

**Rating:** 6
**Confidence:** 3

**Summary:**

The paper derives bounds for the Bayesian simple regret of two GP optimization policies that are variants of the popular expected improvement and UCB, specifically under the assumption that the querying budget is very low compared to the size of the search space.
The authors first do this in the discrete setting where there are a finite number of actions to be taken and then extend their results to the continuous setting.
The bounds are derived using a procedure inspired by the adaptive submodularity framework.
Results show that these new bounds are lower than those from a previous work.

**Limitations And Societal Impact:**

Yes.

**Main Review:**

The paper in general is clear and does a good job setting up the problem and their own approach.
The low-budget assumption is natural in many applications of Bayesian optimization where making a measurement is expensive; this addresses the problem of many of previous work where regret bounds are only meaningful when the budget is large.
The high-level idea for the proof makes intuitive sense: if we are far away from the global optimum, we are more likely to be able to make significant progress at the next iteration.
The included visualizations clearly show that the bounds are much lower than those from Grunewalder et al. (presumably the state-of-the-art bounds in this setting).
Also, the authors are very clear about the limitations of this submission.

My questions for the authors are as follows.

As mentioned in the paper, EI2 and UCB2, for which the bounds are derived, are special variants of EI and UCB that aim to find both the global minimum and the maximum of the function simultaneously.
This symmetry is necessary as it plays a role in the proof for the bounds.
So, these bounds don't apply to EI and UCB, which are commonly used in practice, although the authors think they might, but showing this is not trivial.
I wonder if there are any intermediate results supporting the claim that EI/UCB tends to perform better than EI2/UCB2, as the former only needs to dedicate its budget to one direction of the optimization.
Since the regret of EI2/UCB2 is upper-bounded as established by this paper, EI/UCB may then have the same guarantee?
Something along these lines would make the results more relevant.

A big question I have is about the optimal policy: it is mentioned several times in the paper that the optimal strategy is random sampling without replacement, but I do not see why this is the case.
Isn't the optimal strategy the one found by solving the intractable POMDP, of which EI is the one-step approximation, as mentioned in section 3.3?
It seems to me random sampling is very far from being optimal.

How does Lemma 2 fit in with the rest of the results?
Does it say that despite the bounds derived, there always exist situations where the regret is greater than $1 - T / N$?
If so, could stronger bounds be derived when we place restrinctions on the covariance function like in the continuous case?

A few minor details:
- There is a typo in the notation for the function minimum on line 134.
- It seems equations (22) and (24) have the wrong sign for $\epsilon$.
- EI in equation (10) seems to be defined as a function of the current incumbent $\tau$.
I suggest following the convention and define it as a function of a candidate query/action.

Overall, I think the submission tackles an important problem, but more results may be needed to warrant publication.

**After the authors' response**: Thank you for the detailed comments and answers. I am happy with them and have decided to increase my score to 6. I look forward to seeing the (weaker) bounds for the standard EI/UCB and empirical comparisons between the versions being added to the revision.

**Time Spent Reviewing:**

3

---

> ### Author Response · Authors · 2021-08-10
> **Response**
>
> Thank you for taking the time to read the paper and giving helpful feedback which we will take into account for a potential final version of the paper!
>
> In the following we will address the main points of the review in individual comments to facilitate discussion.

---

> ### Author Response · Authors · 2021-08-10
> **Relation to Standard EI/UCB**
>
> Intuitively, we would expect standard EI/UCB to perform better than
> EI2/UCB2 for maximization: At any given step, evaluating at a potential
> maximizer instead of a minimizer will clearly lead to a larger immediate
> reduction in regret (i.e. for the last optimization step, it would
> be clearly the better choice). However, what is hard to quantify is
> how the information acquired by this evaluation will influence the
> future behavior of the algorithm. Intuitively, we would expect that
> an evaluation at a potential minimizer will not provide any more useful
> information than an evaluation at a potential maximizer. If we could
> prove this to be the case, this would imply that EI/UCB perform better
> than EI2/UCB2 for maximization, and hence that our bounds apply to
> EI/UCB. Unfortunately, proving this formally appears to be surprisingly
> difficult.
>
> Nevertheless, we are able to give regret bounds for standard EI/UCB
> with some small modifications to the proof (we are happy to go into
> more detail, if desired). We did not include these bounds in the paper
> because they are weaker than the ones we derived for EI2/UCB2, but
> we discuss them in the following and intend to include them into a
> final version of the paper. In addition, we will add an empirical
> comparison between EI/UCB and EI2/UCB2.
>
> We obtain the following version of Theorem 1 for standard EI/UCB:
>
> For any problem instance $(N,T,\mu,\Sigma)$ with $N\ge T\ge500$,
> if we follow either the $\text{\textbf{standard EI or UCB strategy}}$
> we have
> \begin{equation}
> \frac{\mathbb{E}\left[\hat{\check{F}}\right]-\mathbb{E}\left[\boldsymbol{\hat{Y}-\check{F}}\right]}{\mathbb{E}\left[\hat{\check{F}}\right]}\le1-\left(1-T^{-\frac{1}{2\sqrt{\pi}}}\right)\sqrt{\frac{\log(T)-\log\left(3\log^{\frac{3}{2}}(T)\right)}{\log(N)}}.
> \end{equation}
> Note that we marked the changes in bold. Now, instead of a guarantee
> on $\hat{\check{Y}}$, we provide a guarantee on $\hat{Y}-\check{F}$,
> i.e. the difference between the best obtained value and the function
> minimum. We can then derive from that a version of Corollary 1:
>
> For any problem instance $(N,T,\mu,\Sigma)$ with zero mean $\mu_{n}=0~\forall n$
> and $N\ge T\ge500$, if we follow either the $\text{\textbf{standard EI or UCB strategy}}$,
> we have
> \begin{equation}
> normreg\le\boldsymbol{2}\left(1-\left(1-T^{-\frac{1}{2\sqrt{\pi}}}\right)\sqrt{\frac{\log(T)-\log\left(3\log^{\frac{3}{2}}(T)\right)}{\log(N)}}\right).
> \end{equation}
> The important thing to note here is the appearance of the factor 2
> in the bound. This means that asymptotically we have
> \begin{equation}
> normreg\le\boldsymbol{2}\left(1-\frac{\sqrt{\log T}}{\sqrt{\log N}}\right)
> \end{equation}
> and
> \begin{equation}
> T\le N^{(1-normreg/\boldsymbol{2})}
> \end{equation}
> which is weaker compared to equations (21),(22). Nevertheless, this
> result sheds some light on the applicability of the analysis in this
> paper to standard EI/UCB and we will include it in a potential final
> version of the paper.

---

> ### Author Response · Authors · 2021-08-10
> **Random sampling vs the optimal policy**
>
> Indeed, the optimal policy is the one found by solving the intractable POMDP, and random sampling is generally far from optimal. However, for the specific case defined in Lemma 1 (i.i.d. arms) uniform sampling without replacement turns out to be optimal, since all arms are equivalent and they do not contain any information about each other (see Appendix C for details). The point of Lemma 1 is to derive a lower bound on the regret, i.e. to show that there exists a problem instance (GP with $\mu=0$ and $\Sigma=I$) where even the optimal policy incurs the regret in equation (23).
> We will be more explicit about this point in a potential final version of the paper.

---

> ### Author Response · Authors · 2021-08-10
> **Regarding Lemma 2**
>
> The point of this result is to show that the regret bounds we derive are nontrivial in the sense that they could not be attained by a naive policy which does not take the mean and covariance of the GP into account (such as e.g. uniform random sampling, which will only perform well in the special case of i.i.d. arms). Note that this lower bound does not apply to EI/UCB and EI2/UCB2, since these policies do take the mean and covariance into account for computing the posterior GP at each time step, which is then used for selecting the next evaluation point (by optimizing an acquisition function).

---

### Decision · Program_Chairs · 2021-09-27

**Decision:**

Accept (Poster)

**Comment:**

This manuscript considers the development of insightful regret bounds for Bayesian optimization when the domain is so large that we have effectively no hope of finding the global optimum. The analysis is novel and the results give insight into performance in this realistic setting.

After a long discussion and continued engagement with the authors, there is consensus among the reviewers that the work is of high quality, relevant to the NeurIPS audience, and significant.

There is also consensus that there is room for improvement in the current manuscript before publication, and I strongly recommend that the authors take the reviewers' comments and suggestions into account when preparing their camera-ready version. In particular, the empirical evaluation designed and carried out during the rebuttal phase should not be lost.